# Linear Time Approximation Algorithm for Column Subset Selection with Local Search

**Yuanbin Zou**[1,2]**, Ziyun Huang**[3]**, Jinhui Xu**[4]**, Jianxin Wang**[1,2,5]**, Qilong Feng**[1,2,*]

[1]School of Computer Science and Engineering, Central South University,
Changsha 410083, China
[2]Xiangjiang Laboratory, Changsha 410205, China
[3]Department of Computer Science and Software Engineering, Penn State Erie,
The Behrend College
[4]Department of Computer Science and Engineering, State University of New York at Buffalo,
NY, USA
[5]The Hunan Provincial Key Lab of Bioinformatics, Central South University,
Changsha 410083, China
`yuanbinzou@csu.edu.cn, csufeng@mail.csu.edu.cn,`
`zxh201@psu.edu, jinhui@buffalo.edu, jxwang@mail.csu.edu.cn`

## Abstract

The Column Subset Selection (CSS) problem has been widely studied in dimensionality reduction and feature selection. The goal of the CSS problem is to output a submatrix $S$, consisting of $k$ columns from an $n \times d$ input matrix $A$ that minimizes the residual error $\|A - SS^{\dagger}A\|_F^2$, where $S^{\dagger}$ is the Moore-Penrose inverse matrix of $S$. Many previous approximation algorithms have non-linear running times in both $n$ and $d$, while the existing linear-time algorithms have a relatively larger approximation ratios. Additionally, the local search algorithms in existing results for solving the CSS problem are heuristic. To achieve linear running time while maintaining better approximation using a local search strategy, we propose a local search-based approximation algorithm for the CSS problem with exactly $k$ columns selected. A key challenge in achieving linear running time with the local search strategy is how to avoid exhaustive enumerations of candidate columns for constructing swap pairs in each local search step. To address this issue, we propose a two-step mixed sampling method that reduces the number of enumerations for swap pair construction from $O(dk)$ to $k$ in linear time. Although the two-step mixed sampling method reduces the search space of local search strategy, bounding the residual error after swaps is a non-trivial task. To estimate the changes in residual error after swaps, we propose a matched swap pair construction method to bound the approximation loss, ensuring a constant probability of loss reduction in each local search step. In expectation, these techniques enable us to obtain the local search algorithm for the CSS problem with theoretical guarantees, where a $53(k + 1)$-approximate solution can be obtained in linear running time $O(ndk^4 \log k)$. Empirical experiments show that our proposed algorithm achieves better quality and time compared to previous algorithms on both small and large datasets. Moreover, it is at least 10 times faster than state-of-the-art algorithms across all large-scale datasets.

---

[*]Corresponding Author

# 1   Introduction

In machine learning, handling high-dimensional datasets often requires the use of dimensionality reduction techniques, among which Singular Value Decomposition (SVD) is one of the most commonly utilized techniques in practice. The goal of SVD is to minimize the Frobenius norm of the error, aiming to achieve a low-rank approximation of a matrix with theoretical guarantees. An alternative way is to select a small subset of columns from the matrix as representations to well approximate the given matrix, which is known as the Column Subset Selection (CSS) problem. The CSS problem has been widely studied in machine learning for dimensionality reduction with improved interpretability. As pointed out in [4, 27, 17, 13, 20], the number of selected columns $k$ is much smaller than both $n$ and $d$, i.e., $k \ll \min\{n, d\}$. We consider the following CSS problem.

**Definition 1.1.** *Given a matrix $A \in \mathbb{R}^{n \times d}$ and a positive integer $k$, the goal of CSS problem is to select $k$ columns of $A$ forming a matrix $S \in \mathbb{R}^{n \times k}$ that minimizes the residual error*

$$\|A - SS^\dagger A\|_F^2,$$

*where $S^\dagger$ represents the Moore-Penrose inverse matrix of S, and $\|A\|_F^2 = \sum_{i=1}^n \sum_{j=1}^d A_{ij}^2$ denotes the square of Frobenius norm.*

The CSS problem is known to be UG-hard [9]. For the CSS problem, several heuristic algorithms [2, 22, 23] were proposed. However, a main concern for these heuristics is the lack of theoretical guarantees on both the running time and approximation error. Several $(1 + \epsilon)$-approximation bi-criteria algorithms have been proposed for the CSS problem [19, 4, 13, 15]. Many of these algorithms achieve running time of $O(nd \cdot \text{poly}(k))$, which is referred to as linear in both $n$ and $d$ [26, 11]. Although these algorithms achieve $(1+\epsilon)$-approximation, they require selecting more than $k$ columns. For the CSS problem with exactly $k$ columns selected, Boutsidis et al. [5] proposed an $O(k^2 \log k)$-approximation algorithm with $O(\min\{n^2 d, nd^2\})$ running time, using leverage score sampling and QR decomposition methods. Deshpande and Rademacher [13] proposed a volume sampling algorithm, which yields a $(k + 1)$-approximation in time $O(kdn^3 \log n)$. Guruswami and Sinop [19] gave an improved approximation algorithm with $O(n^2 dk)$ running time using fast volume sampling method, achieving the same $(k + 1)$-approximation. The running time of these algorithms has at least a quadratic dependence on $n$ or $d$. Deshpande and Vempala [15] presented a linear-time algorithm with $(k + 1)!$-approximation using adaptive sampling method.

Although the algorithm using the adaptive sampling method [15] achieves the linear running time, its approximation ratio is considerably larger than other algorithms. Moreover, as shown in [23], local search can improve the quality of the solution for the CSS problem. However, a potential limitation of the local search algorithm proposed in [23] is the lack of theoretical guarantees on the number of local search steps required for reaching a convergence.

To apply the local search strategy to handle the CSS problem, the running time of each iteration is quadratically dependent on $d$, making it impractical for large-scale datasets. More specifically, the single-swap strategy in local search enumerates $O(kd)$ swaps to improve the current solution, where the swap pair is constructed between the given matrix and the set of selected columns during the local search step. This method results in an $O(nd^2 k^2)$ running time in each iteration, making it difficult to maintain linear running time in both $n$ and $d$. Secondly, to the best of our knowledge, there is no available result that provides an approximation guarantee for solving the CSS problem using the local search strategy.

## 1.1   Our Contribution

In this paper, we propose a local search algorithm for solving the CSS problem with running time linear in both $n$ and $d$. The key challenge for the local search algorithm is to avoid the $O(nd^2 k^2)$ running time caused by enumerating all possible swap pairs. To overcome this challenge, we propose a two-step mixed sampling method that selects a candidate column with a specific probability for swapping, reducing the enumeration of swap pairs from $O(dk)$ to $k$. By applying the two-step mixed sampling method, the running time of each local search step is reduced from $O(nd^2 k^2)$ to $O(ndk^2)$.

Although the two-step mixed sampling method accelerates the local search process, bounding the residual error of the solution after swaps is a non-trivial task. Specifically, it is challenging to theoretically estimate the improvement in the approximation loss of the current solution after swaps.

To address this issue, we propose a matched swap-pair construction method. This method identifies matched column pairs between the current solution and the optimal solution. Based on these matched pairs, we guarantee that the two-step mixed sampling method can find a column. With constant probability, this column reduces the residual error of the updated solution by a multiplicative factor of $1 - \Theta(1/k)$ in each local search step. With this approach, the expected number of iterations can be bounded by $O(k^2 \log k)$. Therefore, we obtain a local search-based approximation algorithm with $O(ndk^4 \log k)$ running time. The main contributions of this paper are summarized as follows.

- For the CSS problem, we propose a new algorithm that uses local search with a two-step mixed sampling method. This method avoids the quadratic dependence of $d$ on running time by reducing the number of swap pair enumerations during the local search process. Additionally, we propose a matched swap-pair construction method to bound the improvement of residual error during swaps. With these techniques, we achieve a $53(k+1)$-approximation with exactly $k$ columns selected, where the running time of our proposed algorithm is $O(ndk^4 \log k)$.

- Numerical experiments show that our algorithm performs better in terms of quality on both small and large datasets compared to previous algorithms that selects exactly $k$ columns, and is at least 10 times faster than the state-of-the-art algorithms on all large datasets.

Table 1: Comparison of existing results for the CSS Problem with exactly $k$ columns selected, where $n$ is the number of rows in the given matrix, $d$ is the number of columns, and $k$ is the number of selected columns.

| References | Approximation Ratio | Method | Running Time |
|---|---|---|---|
| [15] | $(k+1)!$ | adaptive sampling | $O(ndk^2)$ |
| [5] | $O(k^2 \log k)$ | leverage score sampling + QR decomposition | $O(\min\{n^2 d, nd^2\})$ |
| [13] | $k+1$ | volume sampling | $O(kn^3 d \log n)$ |
| [19] | $k+1$ | fast volume sampling | $O(n^2 dk)$ |
| This paper | $53(k+1)$ | local search + two-step mixed sampling | $O(ndk^4 \log k)$ |

## 1.2 Related Work

Within the framework of rank-revealing QR factorization (RRQR) [6], several $\text{poly}(k, d)$-approximation results [3, 7, 18] have been proposed. These results achieve running time of $O(nd^2)$ while selecting exactly $k$ columns to solve the CSS problem with Frobenius norm error. Deshpande et al. [14] gave the lower bound of $(k+1)$-approximation for the problem. Furthermore, Boutsidis et al. [5] proposed a randomized algorithm with $O(\min\{n^2 d, nd^2\})$ running time and $O(k^2 \log k)$-approximation. More precisely, Deshpande and Rademacher [13] provided a deterministic $(k+1)$-approximation with $O(dn^3 \log nk)$ running time. To improve the running time, Guruswami and Sinop [19] proposed an $O(n^2 dk)$ time randomized algorithm with $(k+1)$-approximation. Additionally, there are many bi-criteria algorithms that relax the number of selected columns. Volume sampling methods have been widely applied to the CSS problem. Deshpande and Vempala [15] utilized these methods to achieve a PTAS, selecting $O(k/\epsilon^2 + k^2 \log k)$ columns. Boutsidis et al. [4] proposed a linear-time algorithm with a $(1 + \epsilon)$-approximation, requiring $O(k/\epsilon)$ columns selected. Guruswami and Sinop [19] developed a deterministic algorithm that also achieves a $(1 + \epsilon)$-approximation with $O(k/\epsilon)$ columns selected. Civril and Magdon-Ismail [10] gave improved bounds for obtaining a PTAS using $k$-leverage score sampling and SVD, by selecting $\widetilde{O}(k \log k/\epsilon^2)$ [2] columns. Altschuler et al. [1] developed a distributed greedy algorithm for the objective $\|SS^\dagger A\|_F^2$. Wang and Singh [28] studied the CSS problem in the missing-data case. Several bi-criteria algorithms have been proposed for the CSS problems with $\ell_1$ norm [12, 21, 25, 8]. In both offline and online settings, Woodruff and Yasuda [29] provided several bi-criteria algorithms for the CSS problem with the $\ell_p$ norm.

## 2 Preliminaries

For any positive integer $n$, let $[n]$ denote the set $\{1, 2, \ldots, n\}$. Given a matrix $A \in \mathbb{R}^{n \times d}$, let $A_{ij}$ be the element in the $i$-th row and the $j$-th column of $A$, and define the Frobenius norm of $A$ as

---

[2] $\tilde{O}(n)$ denotes the asymptotic complexity, ignoring polylogarithmic factors, i.e., terms of the form $\text{poly}(\log n)$.

---

**Algorithm 1** LSCSS

---

**Input:** a matrix $A \in \mathbb{R}^{n \times d}$, an integer $k$, and the number of iterations $T$
**Output:** a submatrix consisting of $k$ columns from $A$

1: Initialize $\mathcal{I} = \emptyset, E = A, B = A$.
2: **for** $t = 1, 2$ **do**
3:     **for** $j \leftarrow 1, 2, \ldots, k$ **do**
4:         Sample a column index $i \in [d]$ with probability $p_i = \|E_{:i}\|_2^2 / \|E\|_F^2$.
5:         Update $\mathcal{I} = \mathcal{I} \cup \{i\}$ and $E = A - A_\mathcal{I} A_\mathcal{I}^\dagger A$.
6:     **end for**
7:     **if** $t$ is equal to 1 **then**
8:         Initialize an $n \times d$ zero matrix $D$, and set each diagonal entry $D_{ii} = \frac{\|A - A_\mathcal{I} A_\mathcal{I}^\dagger A\|_F}{(52 \min\{n,d\}(k+1)!)^{1/2}}$.
9:         Compute $A \leftarrow A + D$ and set $\mathcal{I} = \emptyset$.
10:    **end if**
11: **end for**
12: Compute $A' = B + D$ and set $S = A'_\mathcal{I}$.
13: **for** $i \leftarrow 1, 2, \ldots, T$ **do**
14:    $S \leftarrow \mathrm{LS}(A', k, S)$.
15: **end for**
16: Let $\mathcal{I}$ be the set of column indices of $S$.
17: **return** $A_\mathcal{I}$.

---

**Algorithm 2** LS

---

**Input:** a matrix $A' \in \mathbb{R}^{n \times d}$, an integer $k$, and a matrix $S \in \mathbb{R}^{n \times k}$
**Output:** a submatrix consisting of $k$ columns from $A'$

1: Compute the residual matrix $E = A' - SS^\dagger A'$.
2: Sample a set $C$ of $10k$ column indices from $A'$, where each column index $i$ is picked with probability $\|E_{:i}\|_2^2 / \|E\|_F^2$.
3: Uniformly sample an index $p \in C$.
4: Let $\mathcal{I}$ be the set of the columns indices of $S$ in $A'$.
5: **if** there exists an index $q \in \mathcal{I}$ such that $f(A', A'_{\mathcal{I} \setminus \{q\} \cup \{p\}}) < f(A', S)$ **then**
6:     Find an index $q \in \mathcal{I}$ that minimizes $f(A', A'_{\mathcal{I} \setminus \{q\} \cup \{p\}})$.
7:     $\mathcal{I} = \mathcal{I} \setminus \{q\} \cup \{p\}$.
8: **end if**
9: **return** $A'_\mathcal{I}$.

---

$\|A\|_F^2 = \sum_{i=1}^n \sum_{j=1}^d A_{ij}^2$. Denote $A_{:j}$ as the $j$-th column of $A$, and $A_{i:}$ as the $i$-th row of $A$. Let $A^\top$ be the transpose of $A$ and $A^\dagger$ be the Moore-Penrose inverse of $A$. Given an $n \times d$ matrix $A$, let $\mathcal{I}$ be the set of column indices from $A$, and let $A_\mathcal{I}$ denote the $n \times |\mathcal{I}|$ submatrix of $A$ consisting of the columns corresponding to the indices in $\mathcal{I}$. For a matrix $A$, the linear span of its column vectors is denoted as span $(A)$. For any two $n \times d$ matrices $A$ and $B$, $\|AB\|_F \leq \|A\|_F \|B\|_2$ and $\|AB\|_F \leq \|A\|_2 \|B\|_F$. Given any matrix $A \in \mathbb{R}^{n \times d}$, the singular value decomposition (SVD) of $A$ can be written as $A = \sum_{i=1}^n \sigma_i u_i v_i^\top$, where $\sigma_1 \geq \ldots \geq \sigma_n \geq 0$ are the singular values, $\{u_1, \ldots, u_n\} \subseteq \mathbb{R}^n$ are the left singular vectors, and $\{v_1, \ldots, v_d\} \subseteq \mathbb{R}^d$ are the right singular vectors. Denote $rank(A)$ be the rank of a matrix $A$, which is the number of non-zero singular values of $A$. Moreover, we denote $A_k = \sum_{i=1}^k \sigma_i u_i v_i^\top$ as the best rank-$k$ approximation to $A$ under the Frobenius norm. The spectral norm of $A$, denoted by $\|A\|_2$, is defined as the largest singular value of $A$, i.e., $\|A\|_2 = \sigma_{\max}(A)$. Given a solution $S$ to the CSS problem on matrix $A$, we define the residual error of $S$ as $f(A, S) = \|A - SS^\dagger A\|_F^2$.

## 3   Linear Time Local Search Algorithm for CSS Problem

In this section, we propose a local search approximation algorithm for solving the CSS problem, called LSCSS, which maintains a running time linear in both $n$ and $d$. Directly applying single-swap

local search to solve the CSS problem results in an $O(nd^2k^2)$ running time by enumerating all possible swap pairs. Thus, it is challenging to apply the local search method to solve the CSS problem while maintaining a linear dependence on both $n$ and $d$ in the running time. To avoid $O(nd^2k^2)$ running time in each local search step, we propose a two-step mixed sampling method to reduce the running time from $O(nd^2k^2)$ to $O(ndk^2)$. Although the sampling method reduces the running time by directly using the single-swap local search, analyzing the bound of improvement on the residual error after swaps is a difficult task. To provide a theoretical analysis for the local search step, we propose a matched swap pair construction method to bound the improvement on the residual error during swaps. By carefully analyzing the improvement, we show that our proposed algorithm achieves $53(k+1)$-approximation with $O(ndk^4 \log k)$ running time. The detailed algorithm for the CSS problem is given in Algorithm 1.

The LSCSS algorithm mainly comprises local search and two-stage mixed sampling components. The high-level idea behind our proposed local search is to identify a swap pair that minimizes the residual error in each iteration. The swap pair consists of a column from the input matrix to swap in and a column from the current solution to swap out. By repeating this process, the algorithm produces an updated solution with better quality. Moreover, the two-stage mixed sampling method involves two steps for obtaining a candidate column from the input matrix. Firstly, a set of column indices is constructed by sampling each column with probability proportional to its residual error for the current solution. Then, a column is uniformly selected from the set of candidate indices as the final column to swap in. To ensure that the input matrix for the local search process is full rank, we construct a new matrix $A'$ by adding a small perturbation matrix $D$ to the original matrix $A$ during the initialization, where $D$ is full-rank and has non-zero values only on its diagonal. The full-rank property of $A'$ is used in subsequent analysis.

The LSCSS algorithm begins by obtaining an initial solution $S$ with exactly $k$ columns and constructing a full-rank matrix $A'$ during the initialization (steps 1-12 of Algorithm 1), which achieves a $(k+1)!$-approximate solution on $A'$. We start by initializing an index set $\mathcal{I}$ and setting the matrix $E = A$. A new column index is added to $\mathcal{I}$ by sampling each column index $i$ from $[d]$ with probability proportional to $p_i = \|E_{:i}\|_2^2/\|E\|_F^2$. Then, $E$ is updated as $E = A - A_{\mathcal{I}}A_{\mathcal{I}}^{\dagger}A$. Repeating this process $k$ times, we obtain an initial solution $S = A_{\mathcal{I}}$. To construct a full-rank matrix, we construct an $n \times d$ zero matrix $D$ and compute the parameter $\alpha = \|A - SS^{\dagger}A\|_F/(52 \min\{n,d\}(k+1)!)^{-1/2}$ using the initial solution $S$ and $A$. Each diagonal entry $D_{ii}$ is set to $\alpha$. Since $\text{rank}(A + D) = \text{rank}(D)$, we construct the full-rank matrix $A'$ by adding the full-rank matrix $D$ to the input matrix $A$. To solve the CSS problem on $A'$, we execute steps 3-6 of Algorithm 1 to obtain the solution $S = A'_{\mathcal{I}}$. The detailed process described in steps 1-12 of Algorithm 1 requires $O(ndk^2)$ time.

The local search performed in steps 13-15 of Algorithm 1 plays a crucial role in LSCSS, involving two main steps. Firstly, we compute the matrix $E = A' - SS^{\dagger}A'$ for the current solution $S$. Then, a set $C$ of $10k$ column indices is constructed by sampling each column index $i$ from $[d]$ with probability $p_i = \|E_{:i}\|_2^2/\|E\|_F^2$. Next, a column index $p$ is uniformly selected from $C$, referred to as the "swap-in" column index. Let $\mathcal{I}$ denote the set of column indices of $S$ in $A'$. Subsequently, if there exists an index $q \in I$ such that $f(A', A'_{\mathcal{I}\backslash\{q\}\cup\{p\}}) < f(A', S)$, we choose $q$ as the "swap-out" column index and update the set of indices to $\mathcal{I} = \mathcal{I}\backslash\{q\} \cup \{p\}$. Finally, Algorithm 2 returns the solution $S = A'_{\mathcal{I}}$. After repeating this process $T = O(k^2 \log k)$ times, Algorithm 1 returns the final solution $S$ for the input matrix $A$.

In the following, we explain in more detail how our proposed local search algorithm achieves a $53(k+1)$-approximation for the original matrix $A$. Given an initial solution, the main idea for analyzing the approximation ratio of our algorithm is to bound the improvement on residual error during the swaps in the local search step. To achieve this bound, we propose a matched swap pair construction method that guarantees an improvement in the current solution by swapping one column (Lemma 3.6 and Lemma 3.7). By carefully analyzing the improvement, we show that with constant probability the approximation loss of the current solution can be reduced by a multiplicative factor of $1 - \Theta(1/k)$ in each iteration of the local search algorithm (Lemma 3.8). This implies that after $O(k^2 \log k)$ iterations, we have $\|A' - SS^{\dagger}A'\|_F^2 \leq 26(k+1)\|A' - A'_k\|_F^2$ (Theorem 3.9). Finally, by analyzing the change in the residual error caused by removing matrix $D$ from $A'$, we obtain $\|A' - SS^{\dagger}A'\|_F^2 \leq 53(k+1)\|A - A_k\|_F^2$ in expectation (Lemma 3.10).

We assume that the matrix $A$ has been normalized such that $\|A\|_F^2 = 1/4$. Otherwise, we can normalize each element $A_{ij}$ in $A$ as $A_{ij} = \frac{A_{ij}}{2\|A\|_F}$. Next, we consider a single iteration of Algorithm

2. We assume that the current solution has a high residual error (larger than $25(k+1)\|A' - A'_k\|_F^2$ before executing Algorithm 2 on $A'$. Otherwise, the initial solution $S$ is a $25(k+1)$-approximation for the input matrix $A'$.

Let $S^* = \{s_1^*, \ldots, s_k^*\}$ be the optimal solution with exactly $k$ selected for $A'$, and let $S = \{s_1, \ldots, s_k\}$ be the current solution. We define $\phi(A', S^*, S, s^*) = \arg\min_{s \in S} f(A', S^* \backslash \{s^*\} \cup \{s\})$ as a mapping function that finds $s$ from $S$ such that the residual error $f(A', S^* \backslash \{s^*\} \cup \{s\})$ is minimized. Thus, we say that $s^*$ is captured by $\phi(A', S^*, S, s^*)$. Each column $s^* \in S^*$ is captured by exactly one column from $S$. Let $\mathcal{I}$ denote the set of column indices of $S$ in matrix $A'$. We denote $L$ as the set of columns indices in $S$ that do not capture any optimal columns. We denote $H$ as the set of indices where each column in $S$ captures exactly one optimal column.

The main idea behind the matched swap pair construction method is to analyze the change in residual error caused by swapping an index from set $H$ (or $L$) with an index from a sampled column, using a two-step mixed sampling approach for the current solution $S$. For the column $s_h$ (where $h \in H$) in $S$, $s_h$ captures exactly the column $s_h^*$ of the optimal solution $S^*$, serving as the candidate column for $s_h^*$. If the residual error of swapping $s_h$ to replace $s_h^*$ is large, we prove that with constant probability, sampling a new column can reduce the residual error and update $s_h$. Similarly, for the column $s_l$ (where $l \in L$) in $S$, $s_l$ does not match any optimal column. We also show that, with constant probability, sampling a column from the input matrix $A'$ can reduce the residual error for columns in set $L$. To analyze the improvement in residual error during swaps, we focus on a single swap process, evaluating both the increase in residual error from removing a column $s$ from $S$ and the decrease in residual error from inserting a new column. We give the following definition to measure the change resulting from removing a column.

**Definition 3.1.** *Let $A' \in \mathbb{R}^{n \times d}$ be a full-rank matrix, and let $S$ be a solution on $A'$. Let $\mathcal{I}$ be the set of column indices of $S$. The change in residual error by removing the column $i$ from $\mathcal{I}$ is defined as*

$$\tau(A', S, \mathcal{I} \backslash \{i\}) = f(A', A'_{\mathcal{I} \backslash \{i\}}) - f(A', S).$$

To bound $\tau(A', S, \mathcal{I} \backslash i)$ of solution $S$ on the matrix $A'$, we provide the theoretical guarantee in the following lemma. (Detailed proof of Lemma 3.2 is given in Appendix A.1)

**Lemma 3.2.** *Let $A' \in \mathbb{R}^{n \times d}$ be a full-rank matrix, and let $S$ be a solution on $A'$. Let $\mathcal{I}$ be the set of the column indices in $S$. For $i \in \mathcal{I}$, we have*

$$\tau(A', S, \mathcal{I} \backslash \{i\}) \leq \|A'_{\mathcal{I}} A'^{\dagger}_{\mathcal{I}} A'\|_F^2.$$

To further analyze the bound on $\tau(A', S, \mathcal{I} \backslash \{i\})$, we decompose the projection matrix $A'_{\mathcal{I}} A'^{\dagger}_{\mathcal{I}} A'$ and show that the expected upper bound of $\tau(A', S, \mathcal{I} \backslash \{i\})$ is proportional to $\|A'\|_F^2$. (Detailed proof of Lemma 3.3 is given in Appendix A.1)

**Lemma 3.3.** *Let $A' \in \mathbb{R}^{n \times d}$ be a full-rank matrix, $k$ be a positive integer, and let $\mathcal{I}$ be the set of column indices of $S$ for the CSS problem on $A'$. In expectation, the following inequality holds*

$$\|A'_{\mathcal{I}} A'^{\dagger}_{\mathcal{I}} A'\|_F^2 \leq \frac{k^2}{d^2} \|A'\|_F^2.$$

In the following, we theoretically bound the residual error resulting from adding a candidate column index $p$ to the set $\mathcal{I}$ of column indices in $S$, where $p$ is chosen using the two-step mixed sampling method. (Detailed proof of Lemma 3.4 is given in Appendix A.1)

**Lemma 3.4.** *Let $A' \in \mathbb{R}^{n \times d}$ be a full-rank matrix, $k$ be a positive integer, and let $S$ be a solution with the set $\mathcal{I}$ of column indices in $S$. Let $E = A' - SS^{\dagger} A'$. The column index $p$ is obtained by executing steps 2-3 of Algorithm 2. In expectation, the following inequality holds*

$$f(A', A'_{\mathcal{I} \cup \{p\}}) \leq f_k(A', opt) + \frac{1}{10} f(A', S),$$

*where $f_k(A', opt)$ denotes the best rank-$k$ solution.*

According to the aforementioned mapping function $\phi(\cdot)$, we obtain the subset $H$ from the set $\mathcal{I}$ of column indices and the set $R = \mathcal{I} \backslash H$. By using the matched swap pair construction method, there are two cases for the residual error of the current solution:

1. For the set $H$, where $\sum_{h \in H} f(A', A'_{\mathcal{I} \setminus \{h\}}) > \frac{21}{50} \sum_{i \in \mathcal{I}} f(A', A'_{\mathcal{I} \setminus \{i\}})$.

2. For the set $R = \mathcal{I} \setminus H$, where $\sum_{r \in R} f(A', A'_{\mathcal{I} \setminus \{r\}}) \geq \frac{29}{50} \sum_{i \in \mathcal{I}} f(A', A'_{\mathcal{I} \setminus \{i\}})$.

By Lemma 3.2 and Lemma 3.4, we define the good columns $s_i$ for $i \in \mathcal{I}$ with respect to $S$ as follows.

**Definition 3.5.** *Let $A' \in \mathbb{R}^{n \times d}$ be a full-rank matrix, and let $k$ be a positive integer. Let $S^*$ be the optimal solution with exactly $k$ columns selected, and let $\mathcal{I}^*$ be the set of column indices in $S^*$. Let $S$ be any solution with exactly $k$ columns selected, and let $\mathcal{I}$ be the set of column indices in $S$. A column index $i \in \mathcal{I}$ is called good if*

$$f(A', A'_{\mathcal{I} \setminus \{i\}}) - \tau(A', S, \mathcal{I} \setminus \{i\}) - \tau(A', A'_{\mathcal{I} \cup \{p\}}, (\mathcal{I} \cup \{p\}) \setminus \{i\})$$
$$- \frac{11}{10} \left( f(A', A'_{\mathcal{I}^* \setminus \{i^*\}}) + \frac{1}{10} f(A', S) \right) > \frac{1}{100k} f(A', S),$$

*where $i^* \in \mathcal{I}^*$ is the column index mapped from $i \in \mathcal{I}$ by the function $\phi(\cdot)$, and $p$ is a column index obtained by executing steps 2-3 of Algorithm 2.*

Definition 3.5 estimates the gain from replacing the column $s_h$ with a new column obtained using the two-step sampling method. Next, we argue that if case (1) happens, the sum of residual errors for the good columns is large. (Detailed proof of Lemma 3.6 is given in Appendix A.1)

**Lemma 3.6.** *Let $A' \in \mathbb{R}^{n \times d}$ be a full-rank matrix, $k$ be a positive integer, and let $S$ be the solution to the CSS problem on $A'$. Let $\mathcal{I}$ be the set of column indices in $S$. If $50 \sum_{h \in H} f(A', A'_{\mathcal{I} \setminus \{h\}}) \geq 21 \sum_{i \in \mathcal{I}} f(A', A'_{\mathcal{I} \setminus \{i\}})$ and $f(A', S) \geq 25(k+1) f_k(A', opt)$, we have*

$$\sum_{h \in H, h \text{ is good}} f(A', A'_{\mathcal{I} \setminus \{h\}}) \geq \frac{1}{125} \sum_{i \in \mathcal{I}} f(A', A'_{\mathcal{I} \setminus \{i\}}).$$

Since $R = \mathcal{I} \setminus H$, it holds that $L \subseteq R$. The index set $R$ contains two subsets: $L$ and $R \setminus L$, where the indices in $L$ do not capture any optimal columns according to the mapping function $\phi(\cdot)$ and the indices in $R \setminus L$ capture at least two columns. Similar to case (1), we argue that if case (2) occurs, the sum of residual errors for the good columns is large. (Detailed proof of Lemma 3.7 is given in Appendix A.1)

**Lemma 3.7.** *Let $A' \in \mathbb{R}^{n \times d}$ be a full-rank matrix, $k$ be a positive integer, and let $S$ be a solution for the CSS problem on matrix $A'$. Let $\mathcal{I}$ be the set of column indices in $S$. If $\sum_{r \in R} f(A', A'_{\mathcal{I} \setminus \{r\}}) \geq 29/50 \sum_{i \in \mathcal{I}} f(A', A'_{\mathcal{I} \setminus \{i\}})$ and $f(A', S) \geq 25(k+1) f_k(A', opt)$, we have*

$$\sum_{r \in R, r \text{ is good}} f(A', A'_{\mathcal{I} \setminus \{r\}}) \geq \frac{1}{125} \sum_{i \in \mathcal{I}} f(A', A'_{\mathcal{I} \setminus \{i\}}).$$

In the following, we prove that if the residual error of the current solution is larger than $25(k+1) f_k(A', opt)$, each local search step reduces the residual error by a factor of $1 - \Theta(\frac{1}{k})$ with constant probability. (Detailed proof of Lemma 3.8 is given in Appendix A.1)

**Lemma 3.8.** *Let $A' \in \mathbb{R}^{n \times d}$ and $S \in \mathbb{R}^{k \times d}$ be the input matrices for Algorithm 2, where $k$ is a positive integer and $S$ is the solution of the CSS problem on $A'$. Suppose that $f(A', S) \geq 25(k+1) \cdot f_k(A', opt)$. Then, with probability at least $1/1375$, Algorithm 2 returns a new solution $S'$ with*

$$f(A', S') \leq (1 - 1/(100k)) f(A', S).$$

Subsequently, we prove that the LSCSS algorithm achieves a $26(k+1)$-approximation for $A'$ after $O(k^2 \log k)$ iterations.

**Theorem 3.9.** *Let $A' \in \mathbb{R}^{n \times d}$ be the input matrix obtained in step 12 of Algorithm 1, let $k$ be a positive integer, and let $S$ be the solution returned after executing Algorithm 2 $T = O(k^2 \log k)$ times. Then, it holds that*

$$\mathbb{E}[\|A' - SS^\dagger A'\|_F^2] \leq 26(k+1)\|A' - A'_k\|_F^2,$$

*where $A'_k$ is the best rank-$k$ approximation of $A'$ for the CSS problem. The running time of Algorithm 1 is $O(ndk^4 \log k)$.*

*Proof.* Let $\hat{S}$ denote the submatrix consisting of $k$ columns obtained in step 12 of Algorithm 1. For the initial solution $\hat{S}$, Deshpande and Vempala [15] provide an approximation ratio of $(k+1)!$. Before executing steps 13-15 of Algorithm 1, the residual error of the initial solution $\hat{S}$ is larger than $25(k+1)\|A' - A'_k\|_F^2$. According to Lemma 3.8, with probability $1/1375$, we can reduce the residual error by a multiplicative factor of $(1 - 1/100k)$.

Let $T = O(k^2 \log k)$. We define a random process $\mathcal{P}$ with initial residual error $\|A' - \hat{S}\hat{S}^\dagger A'\|_F^2$ of the solution $\hat{S}$ such that for $T$ iterations of Algorithm 2, it reduces the value of $\|A' - \hat{S}\hat{S}^\dagger A'\|_F^2$ by at least $(1 - \frac{1}{100k})$ with probability $1/1375$, and it increases the final value of $\|A' - \hat{S}\hat{S}^\dagger A'\|_F^2$ by $25(k+1)\|A' - A'_k\|_F^2$. It is obvious that $\mathbb{E}[\|A' - SS^\dagger A'\|_F^2] \le \mathbb{E}[\|A' - \hat{S}\hat{S}^\dagger A'\|_F^2]$. Then, we have

$$\mathbb{E}[\mathcal{P}] = 25(k+1)\|A' - A'_k\|_F^2 + \|A' - \hat{S}\hat{S}^\dagger A'\|_F^2 \cdot \sum_{i=0}^{T} \binom{T}{i} \frac{i}{1375} \frac{1374}{1375}^{T-i} \left(1 - \frac{1}{100k}\right)^i$$

$$\le \|A' - \hat{S}\hat{S}^\dagger A'\|_F^2 \cdot \left(1 - \frac{1}{137500k}\right)^{137500k \log(k+1)!} + 25(k+1)\|A' - A'_k\|_F^2$$

$$\le \frac{\|A' - \hat{S}\hat{S}^\dagger A'\|_F^2}{(k+1)!} + 25(k+1)\|A' - A'_k\|_F^2.$$

This implies that $\mathbb{E}[\|A' - SS^\dagger A'\|_F^2 | \hat{S}] \le \frac{\|A' - \hat{S}\hat{S}^\dagger A'\|_F^2}{(k+1)!} + 25(k+1)\|A' - A'_k\|_F^2$.

Thus, we obtain

$$\mathbb{E}[\|A' - SS^\dagger A'\|_F^2] = \sum_{\hat{S}} \mathbb{E}[\|A' - SS^\dagger A'\|_F^2 | \hat{S}] Pr(\hat{S})$$

$$\le \sum_{\hat{S}} Pr(\hat{S}) \left( \frac{\|A' - \hat{S}\hat{S}^\dagger A'\|_F^2}{(k+1)!} + 25(k+1)\|A' - A'_k\|_F^2 \right)$$

$$\le \frac{E[\|A' - \hat{S}\hat{S}^\dagger A'\|_F^2]}{(k+1)!} + 25(k+1)\|A' - A'_k\|_F^2.$$

Since $\|A' - \hat{S}\hat{S}^\dagger A'\|_F^2 \le (k+1)!\|A' - A'_k\|_F^2$ in expectation, we have $\mathbb{E}[\|A' - SS^\dagger A'\|_F^2] \le 26(k+1)\|A' - A'_k\|_F^2$.

**Running Time Analysis.** In LSCSS algorithm, the process of constructing the initial solution in the steps 2-11 of Algorithm 1 takes $O(ndk^2)$ time. In order to obtain an $O(k+1)$-approximate solution, Algorithm 2 requires $O(k^2 \log k)$ iterations. In each iteration, computing the residual matrix requires $O(ndk)$ time. The steps 4-8 of Algorithm 2 require $O(ndk^2)$ time to recalculate the residual error. Therefore, the overall running time of Algorithm 1 is $O(ndk^4 \log k)$. $\qquad\square$

In the following, we analyze the change in residual error caused by replacing the input matrix $A$ with $A' = A + D$, which leads to the final solution of Algorithm 1 achieving a $53(k+1)$-approximation. (Detailed proof of Lemma 3.10 is given in Appendix A.1)

**Lemma 3.10.** *Let $A \in \mathbb{R}^{n \times d}$ be an input matrix, and let $k$ be a positive integer. Define $D$ as an $n \times d$ matrix with elements*

$$D_{ij} = \begin{cases} \frac{\|A - S_1 S_1^\dagger A\|_F}{(52 \min\{n,d\}(k+1)!)^{1/2}}, & if\ i = j \\ 0, & otherwise \end{cases},$$

*where $S_1$ is obtained by executing the first round of steps 3-6 in Algorithm 1. Let $A' = A + D$. The solution $S_2$ returned by executing Algorithm 2 for $T = O(k^2 \log k)$ iterations satisfies*

$$\mathbb{E}[\|A' - S_2 S_2^\dagger A'\|_F^2] \le 53(k+1)\|A - A_k\|_F^2,$$

*where $A_k$ is the best rank-k approximation for $A$.*

# 4 Experiments

In this section, we compare our algorithm for the CSS problem with the previous ones. For hardware, all the experiments are conducted on a machine with 72 Intel Xeon Gold 6230 CPUs and 2TB memory.

**Datasets.** In this paper, we evaluate the performance of our algorithms on a total of 22 real-world datasets. In previous studies [23, 1], the CSS problem typically involves datasets with no more than 100,000 rows and 20,000 columns. We include the 14 smaller datasets listed in Table 5 (Appendix A.2). To extend the evaluation to larger datasets, we include 8 additional datasets detailed in Table 2. Six datasets contain between 40,000 and 480,000 columns, and two contain 400,000 and 8 million rows, respectively. All datasets can be found on the website[345].

**Algorithms and parameters.** In our experimental evaluation, we consider the following five distinct algorithms:

- TwoStage. This is a two-stage algorithm from [5] that combines leverage score sampling and rank-revealing QR factorization.
- Greedy. This is an algorithm in [16, 1], which uses greedy algorithm to generate solution.
- VolumeSampling. This is an algorithm in [13], which uses volume sampling method.
- ILS. This is an algorithm in [23], which uses heuristic local search method.
- LSCSS. This is our algorithm given in Algorithm 1, which uses the two-step mixed sampling and local search methods.

**Methodology** We use the error ratio to evaluate the effectiveness of various algorithms, as defined in [23]. The error ratio is given by the formula $\|A - SS^\dagger A\|_F^2 / \|A - A_k\|_F^2$, where it quantifies the discrepancy between the selected columns and the optimal rank-$k$ matrix approximation. A smaller error ratio indicates better algorithm performance. Following [23], we test the TwoStage, VolumeSampling, ILS, and LSCSS algorithms on each dataset 10 times to calculate the average error ratio and running time. Since the Greedy algorithm is deterministic, it is tested only once per dataset.

Table 2: Summary of the datasets

| Datasets | Instances | Features |
|---|---|---|
| Condition Monitoring of Hydraulic Systems(CMHS) | 2205 | 43680 |
| Farm Ads (FAds) | 4143 | 54877 |
| Electricity Load Diagrams (ELD) | 370 | 140256 |
| Gas | 180 | 150000 |
| YaleB | 16380 | 307200 |
| Twin Gas Sensor Arrays (TGas) | 640 | 480000 |
| Epsilon | 400000 | 2000 |
| Mnist8m | 8000000 | 784 |

**Experimental setup.** For the CSS problem with the Frobenius norm, we run the TwoStage, Greedy, ILS, and LSCSS algorithms on both the 8 large datasets and 14 small datasets, providing the average results for each method. The ILS and our LSCSS algorithm are based on local search method. For fair comparison, we set the number of iterations to be $2k$ for ILS and LSCSS. Since the VolumeSampling requires $O(dkn^3 \log n)$ runtime and $O(n^2 + d^2)$ memory, it cannot handle the 8 large datasets because the algorithm requires more than 48 hours of runtime and over 2TB of memory. However, the other four algorithms generally produce a solution within 48 hours and with less than 2TB of memory. Thus, we only include VolumeSampling in the comparison on the 14 smaller datasets and exclude its results from Tables 3 and 4.

---

[3] https://archive.ics.uci.edu/datasets
[4] https://www.csie.ntu.edu.tw/~cjlin/libsvmtools/datasets/
[5] http://www.cl.cam.ac.uk/research/dtg/attarchive/facedatabase.html.

Table 3: Comparison results on running time for varying $k$ on datasets. If an algorithm fails to output a solution within 48 hours, the running time is set as ">48h". If its memory usage exceeds 2TB, the running time is set as "OOM" (Out of Memory).

| Dataset | k | TwoStage | Greedy | ILS | LSCSS | Dataset | k | TwoStage | Greedy | ILS | LSCSS |
|---|---|---|---|---|---|---|---|---|---|---|---|
| CMHS | 5 | 354.91 | 493.49 | 18.43 | **2.82** | FAds | 5 | 228.66 | 127.31 | 143.08 | **3.40** |
| | 10 | 548.58 | 716.27 | 54.59 | **5.17** | | 10 | 226.53 | 248.89 | 498.29 | **10.58** |
| | 15 | 647.34 | 1045.83 | 122.49 | **13.39** | | 15 | 219.35 | 369.95 | 1032.53 | **17.65** |
| | 20 | 847.87 | 1362.44 | 219.43 | **30.45** | | 20 | 221.97 | 495.09 | 1744.18 | **18.27** |
| | 25 | 1150.13 | 1679.30 | 331.24 | **43.27** | | 25 | 229.24 | 624.69 | 2659.60 | **22.29** |
| | 30 | 1249.27 | 1955.13 | 476.28 | **52.82** | | 30 | 232.34 | 749.99 | 3789.49 | **22.76** |
| | 50 | 1364.90 | 2598.43 | 1321.73 | **62.36** | | 50 | 291.93 | 1265.93 | 10363.85 | **36.04** |
| | 100 | 1552.57 | 7263.06 | 4457.66 | **108.98** | | 100 | 539.00 | 2738.69 | 44037.65 | **95.67** |
| ELD | 5 | 434.60 | 245.14 | 25.28 | **2.18** | TGas | 5 | 822.75 | 312.98 | 154.37 | **17.15** |
| | 10 | 539.22 | 436.11 | 50.98 | **3.47** | | 10 | 1205.25 | 617.40 | 539.61 | **30.28** |
| | 15 | 740.71 | 621.19 | 97.00 | **5.36** | | 15 | 1686.43 | 960.75 | 1202.76 | **36.99** |
| | 20 | 940.42 | 1598.41 | 148.07 | **13.93** | | 20 | 2295.88 | 1335.77 | 2095.39 | **49.90** |
| | 25 | 1237.54 | 1710.38 | 236.48 | **22.98** | | 25 | 2976.99 | 1767.18 | 3349.04 | **61.36** |
| | 30 | 1536.68 | 2307.40 | 291.32 | **35.03** | | 30 | 3818.48 | 2230.84 | 4760.69 | **77.93** |
| | 50 | 1738.76 | 3649.70 | 727.45 | **66.27** | | 50 | 7098.36 | 4419.23 | 6772.58 | **114.23** |
| | 100 | 2384.16 | 5946.40 | 2696.24 | **89.94** | | 100 | 9186.20 | 12439.81 | 9352.80 | **243.89** |
| Gas | 5 | 404.52 | 426.89 | 56.71 | **5.83** | YaleB | 5 | OOM | 17214.54 | 2190.74 | **175.24** |
| | 10 | 597.72 | 785.37 | 148.36 | **11.67** | | 10 | OOM | 21210.58 | 9855.42 | **501.67** |
| | 15 | 858.04 | 1162.98 | 202.17 | **19.30** | | 15 | OOM | >48h | 12754.29 | **793.47** |
| | 20 | 1312.71 | 1546.90 | 428.71 | **47.90** | | 20 | OOM | >48h | 23694.02 | **1082.21** |
| | 25 | 1741.95 | 1977.13 | 1623.66 | **130.11** | | 25 | OOM | >48h | 35259.86 | **1756.18** |
| | 30 | 2237.17 | 3317.12 | 4858.71 | **322.92** | | 30 | OOM | >48h | >48h | **4108.82** |
| | 50 | 4944.07 | 7852.03 | 7888.39 | **386.15** | | 50 | OOM | >48h | >48h | **7127.35** |
| | 100 | 20955.39 | 12703.14 | 12190.91 | **850.01** | | 100 | OOM | >48h | >48h | **13865.08** |
| Mnist8m | 5 | 18122.53 | 5564.12 | 3255.38 | **483.02** | Epsilon | 5 | 8916.54 | 1067.48 | 283.22 | **24.85** |
| | 10 | 22216.92 | 34765.28 | 21587.49 | **2991.29** | | 10 | 9221.10 | 2111.57 | 979.81 | **38.71** |
| | 15 | >48h | >48h | 53973.46 | **3420.13** | | 15 | 9426.96 | 3193.27 | 2093.51 | **69.00** |
| | 20 | >48h | >48h | >48h | **6609.07** | | 20 | 10330.94 | 4332.60 | 3645.36 | **102.46** |
| | 25 | >48h | >48h | >48h | **8485.13** | | 25 | 10928.29 | 5570.65 | 5564.13 | **163.25** |
| | 30 | >48h | >48h | >48h | **9495.18** | | 30 | 11371.33 | 6861.29 | 7727.87 | **352.39** |
| | 50 | >48h | >48h | >48h | **13236.29** | | 50 | 13839.09 | 12451.47 | 16939.53 | **562.20** |
| | 100 | >48h | >48h | >48h | **16910.19** | | 100 | 15334.64 | 29352.57 | 18239.98 | **1102.80** |

**Results for the CSS problem.** Table 3 shows the comparison of running time for varying values of $k$, where the time is measured by seconds. LSCSS is at least 10 times faster than other algorithms across all datasets and at least 15 times faster than the TwoStage and Greedy algorithms. Our algorithm successfully outputs a feasible solution within 5 hours on all datasets, whereas other algorithms fail to do so within 48 hours or need more than 2TB memory. The comparison of error ratios, reported as mean±std with the best results highlighted in bold, is presented in Table 4 in the Appendix A.2. The LSCSS algorithm achieves the best error ratios on almost all datasets.

Moreover, we compare the running time and error ratio of five algorithms with varying values of $k$ on 14 small datasets (Appendix A.2). The experimental results show that the LSCSS algorithm outperforms other algorithms in terms of quality and is at least 2 times faster than Greedy, VolumeSampling and ILS algorithms on all small datasets.

# 5  Conclusion

In this paper, we propose a linear-time approximation algorithm for the CSS problem using local search and two-step mixed sampling methods. Experimental results demonstrate that our framework outperforms previous algorithms for solving the CSS problem with exactly $k$ columns selected. An interesting future direction is how to design multi-swap local search approximation algorithms for handling the CSS problem.

# Acknowledgments

This work was supported by National Natural Science Foundation of China (62432016, 62172446), Open Project of Xiangjiang Laboratory (22XJ02002), and Central South University Research Programme of Advanced Interdisciplinary Studies (2023QYJC023). This work was also carried out in part using computing resources at the High Performance Computing Center of Central South University.

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

# A    Appendix / supplemental material

## A.1    Missing Proofs

**Lemma 3.2.** *Let $A' \in \mathbb{R}^{n \times d}$ be a full-rank matrix, and let $S$ be a solution on $A'$. Let $\mathcal{I}$ be the set of column indices in $S$. For $i \in \mathcal{I}$, we have*

$$\tau(A', S, \mathcal{I} \setminus \{i\}) \leq \|A'_{\mathcal{I}} A'^\dagger_{\mathcal{I}} A'\|_F^2.$$

*Proof.* Given a solution $S$, we demonstrate an equivalent transformation to simplify the expression of the residual error for $S$:

$$
\begin{aligned}
\|A' - SS^\dagger A'\|_F^2 &= \mathrm{tr}((A' - SS^\dagger A')^\top (A' - SS^\dagger A')) \\
&= \mathrm{tr}(A'^\top A' - A'^\top SS^\dagger A' - A'^\top (SS^\dagger)^\top A' + A'^\top (SS^\dagger)^\top SS^\dagger A') \\
&= \mathrm{tr}(A'^\top A' - 2A'^\top SS^\dagger A' + A'^\top SS^\dagger A') \\
&= \mathrm{tr}(A'^\top A') - \mathrm{tr}(A'^\top SS^\dagger A') \\
&= \mathrm{tr}(A'^\top A') - \mathrm{tr}(A'^\top (SS^\dagger)^\top SS^\dagger A') \\
&= \mathrm{tr}(A'^\top A') - \|SS^\dagger A'\|_F^2,
\end{aligned}
$$

where the third equality follows from $S^\dagger SS^\dagger = S^\dagger$ and $(SS^\dagger)^\top = SS^\dagger$.

By the above equality, we have

$$
\begin{aligned}
f(A', S') - f(A', S) &= \|A' - S_{\mathcal{I} \setminus \{i\}} S^\dagger_{\mathcal{I} \setminus \{i\}} A'\|_F^2 - \|A' - S_{\mathcal{I}} S^\dagger_{\mathcal{I}} A'\|_F^2 \\
&= \|S_{\mathcal{I}} S^\dagger_{\mathcal{I}} A'\|_F^2 - \|S_{\mathcal{I} \setminus \{i\}} S^\dagger_{\mathcal{I} \setminus \{i\}} A'\|_F^2 \\
&\leq \|S_{\mathcal{I}} S^\dagger_{\mathcal{I}} A'\|_F^2.
\end{aligned}
$$

$\square$

**Lemma 3.3.** *Let $A' \in \mathbb{R}^{n \times d}$ be a full-rank matrix, $k$ be a positive integer, and let $\mathcal{I}$ be the set of column indices of $S$ for the CSS problem on $A'$. In expectation, the following inequality holds:*

$$\|A'_{\mathcal{I}} A'^\dagger_{\mathcal{I}} A'\|_F^2 \leq \frac{k^2}{d^2} \|A'\|_F^2.$$

*Proof.* We begin by seeking to bound the Frobenius norm of the matrix product $A'_{\mathcal{I}} A'^\dagger_{\mathcal{I}} A'$. Using the submultiplicative property of the Frobenius norm, we get

$$\|A'_{\mathcal{I}} A'^\dagger_{\mathcal{I}} A'\|_F^2 \leq \|A'_{\mathcal{I}} A'^\dagger_{\mathcal{I}}\|_F^2 \cdot \|A'\|_F^2.$$

Next, by the norm inequality, we have

$$\|A'_{\mathcal{I}} A'^\dagger_{\mathcal{I}}\|_F^2 \leq \|A'^\dagger_{\mathcal{I}}\|_2^2 \cdot \|A'_{\mathcal{I}}\|_F^2.$$

Since the matrix $A$ is full rank, which is ensured by steps 7-10 of Algorithm 1. Therefore, $\|A'^\dagger_{\mathcal{I}}\|_2^2$ can be reformulated as the maximum eigenvalue of $A'^\top_{\mathcal{I}} A'_{\mathcal{I}}$. Specifically, we have

$$\|A'^\dagger_{\mathcal{I}}\|_2^2 = \frac{1}{\lambda_{\min}(A'^\top_{\mathcal{I}} A'_{\mathcal{I}})} = \lambda_{\max}(A'^\top_{\mathcal{I}} A'_{\mathcal{I}}).$$

Substituting this into $\|A'_{\mathcal{I}} A'^\dagger_{\mathcal{I}} A'\|_F^2$, we get

$$\|A'_{\mathcal{I}} A'^\dagger_{\mathcal{I}} A'\|_F^2 \leq \lambda_{\max}(A'^\top_{\mathcal{I}} A'_{\mathcal{I}}) \cdot \|A'_{\mathcal{I}}\|_F^2 \cdot \|A'\|_F^2.$$

Since the maximum eigenvalue $\lambda_{\max}(A'^\top_{\mathcal{I}} A'_{\mathcal{I}})$ is bounded above by $\|A'_{\mathcal{I}}\|_F^2$, we obtain

$$\|A'_{\mathcal{I}} A'^\dagger_{\mathcal{I}} A'\|_F^2 \leq \|A'_{\mathcal{I}}\|_F^4 \cdot \|A'\|_F^2.$$

Finally, by the expectation $\mathbb{E}[\|A'_{\mathcal{I}}\|_F^2] = \frac{k}{d}\|A'\|_F^2$ and $\|A\|_F^2 = \frac{1}{4}$, we have

$$
\begin{aligned}
\|A'_{\mathcal{I}}\|_F^4 &\leq \left(\frac{k}{d}\right)^2 \|A'\|_F^4 \\
&= \frac{k^2}{d^2}(\|A+D\|_F^2)^2 \\
&\leq \frac{k^2}{d^2}(2\|A\|_F^2 + 2\|D\|_F^2)^2 \\
&= \frac{k^2}{d^2}(2\|A\|_F^2 + 2\sum_{i=1}^{\min\{n,d\}} D_{ii}^2)^2 \\
&= \frac{k^2}{d^2}(2\|A\|_F^2 + 2\min\{n,d\}\frac{\|A-SS^\dagger A\|_F^2}{52\min\{n,d\}(k+1)!})^2 \\
&\leq \frac{k^2}{d^2}(2\|A\|_F^2 + 2\frac{\|A\|_F^2}{52(k+1)!})^2 \\
&\leq \frac{k^2}{d^2}(4\|A\|_F^2)^2 \\
&= \frac{k^2}{d^2}.
\end{aligned}
$$

where the first equality results from $A' = A + D$, and the second inequality follows from the property $\|A+B\|_F^2 \leq 2(\|A\|_F^2 + \|B\|_F^2)$ for any two matrices $A, B \in \mathbb{R}^{n\times d}$.

Combining the above inequalities, we conclude that

$$
\|A'_{\mathcal{I}}A'^\dagger_{\mathcal{I}}A'\|_F^2 \leq \frac{k^2}{d^2}\|A'\|_F^2.
$$

$\square$

**Lemma 3.4.** *Let $A' \in \mathbb{R}^{n\times d}$ be a full-rank matrix, $k$ be a positive integer, and let $S$ be a solution with the set $\mathcal{I}$ of column indices in $S$. Let $E = A' - SS^\dagger A'$. The column index $p$ is obtained by executing steps 2-3 of Algorithm 2. In expectation, the following inequality holds*

$$
f(A', A'_{\mathcal{I}\cup\{p\}}) \leq f_k(A', opt) + \frac{1}{10}f(A', S),
$$

*where $f_k(A', opt)$ denotes the best rank-k solution.*

*Proof.* Let $p_j = \frac{\|E_{:j}\|_F^2}{\|E\|_F^2}$ be the probability that each column index $j \in [d]$ is sampled. To bound $f(A, A_{T\cup\{p\}})$, we define a random variable $X_i^l$ for $i \in [r]$, $j \in [d]$, and $l \in [z]$ as follows:

$$
X_i^l = \frac{v_{ij}}{p_j}E_{:j} \text{ with probability } p_j.
$$

We denote $X_i$ as the random variable that is randomly picked from the set $C = \{X_i^1, X_i^2, \ldots, X_i^l\}$. Therefore, the expected value of $X_i$ is given by

$$
\begin{aligned}
\mathbb{E}(X_i) &= \sum_{l=1}^{z} \mathbf{Pr}(l\text{-th is picked}) \cdot \mathbb{E}\left(X_i^l\right) \\
&= \frac{1}{z}\sum_{l=1}^{z}\mathbb{E}(X_i^l) \\
&= \frac{1}{z}\sum_{l=1}^{z}\sum_{j=1}^{d}\left(p_j \cdot \frac{v_{ij}}{p_j}E_{:j}\right)
\end{aligned}
$$

$$= \frac{1}{z} \sum_{l=1}^{z} \sum_{j=1}^{d} v_{ij} E_{:j}$$

$$= \frac{1}{z} \sum_{l=1}^{z} E v_i$$

$$= E v_i,$$

where the second equality follows from the linearity of expectation, and the last second equality results from the linear combination of $v_i$ and $E$.

Let $w_i = (SS^\dagger A)v_i + X_i$ for $i \in [r]$. Therefore, we have $\mathbb{E}(w_i) = \sigma_i u_i$.

Thus, we have the equality $w_i - \sigma_i v_i = X_i - E v_i$. By calculating its second central moment, we obtain

$$\mathbb{E}(\|w_i - \sigma_i u_i\|_2^2) = \mathbb{E}(\|X_i - E v_i\|_2^2)$$

$$= \mathbb{E}(\|X_i\|_2^2) - 2\mathbb{E}(X_i) \cdot E v_i + \|E v_i\|_2^2$$

$$= \mathbb{E}(\|X_i\|_2^2) - \|E v_i\|_2^2.$$

Next, we seek to bound $\mathbb{E}(\|X_i\|_2^2)$,

$$\mathbb{E}(\|X_i\|_2^2) = \mathbb{E}\left(\|\sum_{l=1}^{z} \mathbf{Pr}(l\text{-th is picked}) \cdot X_i^l\|^2\right)$$

$$= \sum_{l=1}^{z} (\mathbf{Pr}(l\text{-th is picked}))^2 \mathbb{E}(\|X_i^l\|_2^2) + \sum_{1 \le l_1 < l_2 \le z} \mathbf{Pr}(l_1\text{-th and } l_2\text{-th are picked})\mathbb{E}(X_i^{l_1} \cdot X_i^{l_2})$$

$$= \frac{1}{z^2} \sum_{l=1}^{z} \mathbb{E}(\|X_i^l\|_2^2) + \frac{2}{z^2} \sum_{1 \le l_1 < l_2 \le z} \mathbb{E}(X_i^{l_1} \cdot X_i^{l_2})$$

$$= \frac{1}{z^2} \sum_{l=1}^{z} \mathbb{E}(\|X_i^l\|_2^2) + \frac{z-1}{z} \|E v_i\|_2^2.$$

The term of $\frac{2}{z^2} \sum_{1 \le l_1 < l_2 \le z} \mathbb{E}(X_i^{l_1} \cdot X_i^{l_2}) = \frac{z-1}{z}\|E v_i\|_2^2$ follows from the independence of $X_i^{l_1}$ and $X_i^{l_2}$. Therefore, we conclude that

$$\mathbb{E}(\|w_i - \sigma_i u_i\|_2^2) = \frac{1}{z^2} \sum_{l=1}^{z} \mathbb{E}(\|X_i^l\|_2^2) - \frac{1}{z}\|E v_i\|_2^2.$$

For the first term in the above equation, we have

$$\mathbb{E}(\|X_i^l\|_2^2) = \sum_{j=1}^{d} p_j \frac{\|E_{:j} v_{ij}\|_2^2}{p_j^2}$$

$$= \sum_{j=1}^{d} \frac{\|E\|_F^2}{\|E_{:j}\|_F^2} \|E_{:j} v_{ij}\|_2^2$$

$$= \|E\|_F^2 \cdot \sum_{i=1}^{d} \frac{\|E_{:j} v_{ij}\|_F^2}{\|E_{:j}\|_2^2}$$

$$\le \|E\|_F^2 \cdot \frac{\|E v_i\|_2^2}{\|E\|_F^2}$$

$$= \|E v_i\|_2^2.$$

Combining the above inequality, we have

$$\mathbb{E}(\|w_i - \sigma_i u_i\|_2^2) \le \frac{1}{z}\|E v_i\|_2^2.$$

Let $y_i = w_i/\sigma_i$ for $i \in [k]$ and let matrix $F = (\sum_{i=1}^{k} y_i u_i^T) A'$. Therefore, we have $\|A' - A'_{\mathcal{I}\cup\{p\}} A'^{\dagger}_{\mathcal{I}\cup\{p\}} A'\|_F^2 \le \|A' - F\|_F^2$. By decomposing $F$ along the right singular vectors $\{v_1, \ldots, v_d\}$, we have:

$$\mathbb{E}(\|A' - A'_{\mathcal{I}\cup\{p\}} A'^{\dagger}_{\mathcal{I}\cup\{p\}} A'\|_F^2) \le \mathbb{E}(\|A' - F\|_F^2)$$

$$= \sum_{i=1}^{d} \mathbb{E}(\|(A' - F)v_i\|_2^2)$$

$$\le \sum_{i=1}^{k} \mathbb{E}(\|\sigma_i u_i - w_i\|_2^2) + \sum_{i=k+1}^{d} \sigma_i^2$$

$$\le f_k(A', opt) + \frac{k}{z} f(A', A'_{\mathcal{I}}),$$

where the last second inequality uses that $f_k(A', opt) = \sum_{i=k+1}^{d} \sigma_i^2$.

Setting the $z = 10k$, we obtain that

$$f(A', A'_{\mathcal{I}\cup\{p\}}) \le f_k(A', opt) + \frac{1}{10} f(A', S).$$

$\square$

**Lemma 3.6.** *Let $A' \in \mathbb{R}^{n \times d}$ be a full-rank matrix, $k$ be a positive integer, and let $S$ be the solution to the CSS problem on $A'$. Let $\mathcal{I}$ be the set of column indices in $S$. If $50 \sum_{h \in H} f(A', A'_{\mathcal{I}\setminus\{h\}}) \ge 21 \sum_{i \in \mathcal{I}} f(A', A'_{\mathcal{I}\setminus\{i\}})$ and $f(A', S) \ge 25(k+1) f_k(A', opt)$, we have*

$$\sum_{h \in H, h \text{ is good}} f(A', A'_{\mathcal{I}\setminus\{h\}}) \ge \frac{1}{125} \sum_{i \in \mathcal{I}} f(A', A'_{\mathcal{I}\setminus\{i\}}).$$

*Proof.* First, we prove that the residual error is non-increasing. Let $S_k$ be a subset of $k$ columns from $A'$. Let $p \in A'$ be a column not in $S_k$, and define $S_{k+1} = S_k \cup \{p\}$. The residual error function is defined as:

$$f(A', S) = \|A' - SS^{\dagger}A'\|_F^2 = \|A'\|_F^2 - \|SS^{\dagger}A'\|_F^2.$$

Our goal is to show that $f(A', S_{k+1}) \le f(A', S_k)$, i.e., the residual error is non-increasing when a column is added to $S_k$.

Since $\|A'\|_F^2$ is constant, we only need to prove that $\|S_{k+1}S_{k+1}^{\dagger}A'\|_F^2 \ge \|S_k S_k^{\dagger}A'\|_F^2$. For any column $u$ of $A'$, denote

$$\Delta(u) = \|S_{k+1}S_{k+1}^{\dagger}u\|_2^2 - \|S_k S_k^{\dagger}u\|_2^2.$$

If $p \in \text{span}(S_k)$, then $\text{span}(S_{k+1}) = \text{span}(S_k)$, and thus $\Delta(u) = 0$. Otherwise, if $p \notin \text{span}(S_k)$, define the orthogonal component of $p$ with respect to $\text{span}(S_k)$ as

$$p_{\perp} = p - S_k S_k^{\dagger} p,$$

and normalize it as $p' = \frac{p_{\perp}}{\|p_{\perp}\|_2}$.

For any $u$, the projection onto $\text{span}(S_{k+1})$ is $S_{k+1}S_{k+1}^{\dagger}u = S_k S_k^{\dagger}u + \langle u, p'\rangle p'$, where $\langle u, p'\rangle$ denotes the inner product. Therefore, $\|S_{k+1}S_{k+1}^{\dagger}u\|_2^2 = \|S_k S_k^{\dagger}u\|_2^2 + |\langle u, p'\rangle|^2$ holds, implying that $\Delta(u) = |\langle u, p'\rangle|^2 \ge 0$. Summing over all columns $u$ of $A'$, we obtain $\|S_{k+1}S_{k+1}^{\dagger}A'\|_F^2 - \|S_k S_k^{\dagger}A'\|_F^2 = \sum_u \Delta(u) \ge 0$. Thus, the residual error function satisfies

$$f(A', S_{k+1}) = \|A'\|_F^2 - \|S_{k+1}S_{k+1}^{\dagger}A'\|_F^2 \le \|A'\|_F^2 - \|S_k S_k^{\dagger}A'\|_F^2 = f(A', S_k).$$

This proves that adding a column to $S_k$ does not increase the residual error.

By Lemma 3.2, we have

$$\tau(A', A'_{\mathcal{I}\cup\{p\}}, \mathcal{I} \cup \{p\}\setminus\{h\}) = f(A', A'_{\mathcal{I}\cup\{p\}\setminus\{h\}}) - f(A', A'_{\mathcal{I}\cup\{p\}}) \le \|A'_{\mathcal{I}\cup\{p\}} A'^{\dagger}_{\mathcal{I}\cup\{p\}} A'\|_F^2.$$

Similarly, we have

$$\tau(A', S, \mathcal{I}\backslash\{h\}) = f(A', A'_{\mathcal{I}\backslash\{h\}}) - f(A', S) \le \|SS^\dagger A'\|_F^2.$$

Thus, by Lemma 3.3, we obtain $\tau(A', A'_{\mathcal{I}\cup\{p\}}, \mathcal{I}\cup\{p\}\backslash\{h\}) + \tau(A', S, \mathcal{I}\backslash\{h\}) \le \frac{(k+1)^2}{d^2}\|A\|_F^2 + \frac{k^2}{d^2}\|A\|_F^2 \le \frac{2(k+1)^2}{d^2}\|A\|_F^2$, where $h \in H$.

Next, we have $\sum_{h\in H} f(A', S\backslash\{h\}) \ge \frac{21}{50}\sum_{i\in\mathcal{I}} f(A', A'_{\mathcal{I}\backslash\{i\}})$. By Definition 3.5 and Lemma 3.2, we have

$$
\begin{aligned}
\sum_{h\in H, h \text{ is not good}} f(A', A'_{\mathcal{I}\backslash\{h\}}) &\le \sum_{h\in H} \tau(A', S, \mathcal{I}\backslash\{h\}) + \tau(A', A'_{\mathcal{I}\cup\{p\}}, \mathcal{I}\cup\{p\}\backslash\{h\}) \\
&\quad + \frac{11k}{10}(f(A', A'_{\mathcal{I}^*\backslash\{h^*\}}) + \frac{1}{10}f(A', S)) + \frac{1}{100}f(A', S) \\
&\le \frac{2k(k+1)^2}{d^2}\|A'\|_F^2 + \frac{11k}{10}(\frac{k^2}{d^2}\|A'\|_F^2 + f(A', S^*)) \\
&\quad + \frac{11k+1}{100}f(A', S) \\
&\le \frac{31k^3}{10d^2}\|A'\|_F^2 + \frac{11k}{10}f(A', S^*) + \frac{11k+1}{100}f(A', S) \\
&\le \frac{31k^3}{10d^2}\|A'\|_F^2 + \frac{11k(k+1)}{10}f_k(A', opt) + \frac{11k+1}{100}f(A', S) \\
&\le \frac{11k(k+1)}{10}f_k(A', opt) + \frac{11k+1}{100}f(A', S) \\
&\quad + \frac{31k^3}{10d^2}\frac{\|A\|_F^2}{f(A', opt)}f_k(A', opt) \\
&\le \frac{11k(k+1)}{10}f_k(A', opt) + \frac{11k+1}{100}f(A', S) + \frac{31k^3}{5d^2}f_k(A', opt),
\end{aligned}
$$

where the fourth inequality follows from the fact that $S^*$ is the optimal solution with exactly $k$ columns selected, and $f(A', S^*) \le (k+1)f_k(A', opt)$.

Using $f(A', S) \ge 25(k+1)f_k(A', opt)$ and the non-increasing property of residual error, we obtain that

$$\sum_{h\in H, h \text{ is not good}} f(A', A'_{\mathcal{I}\backslash\{h\}}) \le \frac{103k}{250} \cdot f(A', S).$$

By $kf(A', S) \le \sum_{i\in\mathcal{I}} f(A', A'_{\mathcal{I}\backslash\{i\}})$,

$$\sum_{h\in H, h \text{ is not good}} f(A', A'_{\mathcal{I}\backslash\{h\}}) \le \frac{103}{250} \cdot \sum_{i\in\mathcal{I}} f(A', A'_{\mathcal{I}\backslash\{i\}}).$$

Thus, we have

$$\sum_{h\in H, h \text{ is good}} f(A', A'_{\mathcal{I}\backslash\{h\}}) \ge \frac{1}{125} \cdot \sum_{i\in\mathcal{I}} f(A', A'_{\mathcal{I}\backslash\{i\}}).$$

$\square$

**Lemma 3.7.** *Let $A' \in \mathbb{R}^{n\times d}$ be a full-rank matrix, $k$ be a positive integer, and let $S$ be a solution for the CSS problem on matrix $A'$. Let $\mathcal{I}$ be the set of column indices in $S$. If $\sum_{r\in R} f(A', A'_{\mathcal{I}\backslash\{r\}}) \ge 29/50\sum_{i\in\mathcal{I}} f(A', A'_{\mathcal{I}\backslash\{i\}})$ and $f(A', S) \ge 25(k+1)f_k(A', opt)$, we have*

$$\sum_{r\in R, r \text{ is good}} f(A', A'_{\mathcal{I}\backslash\{r\}}) \ge \frac{1}{125} \sum_{i\in\mathcal{I}} f(A', A'_{\mathcal{I}\backslash\{i\}}).$$

*Proof.* By Lemma 3.3, we have $\tau(A', A'_{\mathcal{I}\cup\{p\}}, \mathcal{I}\cup\{p\}\setminus\{l\}) + \tau(A', S, \mathcal{I}\setminus\{l\}) \le \frac{(k+1)^2}{d^2}\|A\|_F^2 + \frac{k^2}{d^2}\|A\|_F^2 \le \frac{2(k+1)^2}{d^2}\|A\|_F^2$, where $l \in L$.

We have $\sum_{r\in R} f(A', A'_{\mathcal{I}\setminus\{r\}}) \ge 29/50 \sum_{i\in\mathcal{I}} f(A', A'_{\mathcal{I}\setminus\{i\}})$. Note that $|R| \le 2|L|$. By Definition 3.5 and Lemma 3.2, we have

$$\sum_{r\in R, r \text{ is not good}} f(A', A'_{\mathcal{I}\setminus\{r\}}) \le 2|L| \min_{l\in L} \tau(A', S, \mathcal{I}\setminus\{l\}) + \tau(A', A'_{\mathcal{I}\cup\{p\}}, \mathcal{I}\cup\{p\}\setminus\{l\})$$

$$+ \frac{11k}{10}(f(A', A'_{\mathcal{I}^*\setminus\{l^*\}}) + \frac{1}{10}f(A', S)) + \frac{1}{100}f(A', S)$$

$$\le 2\sum_{l\in L} \tau(A', S, \mathcal{I}\setminus\{l\}) + \tau(A', A'_{\mathcal{I}\cup\{p\}}, \mathcal{I}\cup\{p\}\setminus\{l\})$$

$$+ \frac{11k}{10}(f(A', A'_{\mathcal{I}^*\setminus\{l^*\}}) + \frac{1}{10}f(A', S)) + \frac{1}{100}f(A', S)$$

$$\le \frac{4k(k+1)^2}{d^2}\|A'\|_F^2 + \frac{11k}{10}(\frac{k^2}{d^2}\|A'\|_F^2 + f(A', S^*))$$

$$+ \frac{11k+1}{100}f(A', S)$$

$$\le \frac{51k(k+1)^2}{10d^2}\|A'\|_F^2$$

$$+ \frac{11k(k+1)}{10}f_k(A', opt) + \frac{11k+1}{100}f(A', S)$$

$$\le \frac{51k^3}{10d^2}\frac{\|A'\|_F^2}{f_k(A', opt)}f_k(A', opt)$$

$$+ \frac{11k(k+1)}{10}f_k(A', opt) + \frac{11k+1}{100}f(A', S)$$

$$\le \frac{51k^3}{5d^2}f_k(A', opt) + \frac{11k(k+1)}{10}f_k(A', opt) + \frac{11k+1}{100}f(A', S).$$

Using $f(A', S) \ge 25(k+1)f_k(A', opt)$ and the non-increasing property of the residual error (as in the proof of Lemma 3.6), we obtain

$$\sum_{r\in R, r \text{ is not good}} f(A', A'_{\mathcal{I}\setminus\{l\}}) \le \frac{143k}{250}f(A', S)$$

.

By $kf(A', S) \le \sum_{i\in\mathcal{I}} f(A', A'_{\mathcal{I}\setminus\{i\}})$, we have

$$\sum_{r\in R, r \text{ is not good}} f(A', A'_{\mathcal{I}\setminus\{l\}}) \le \frac{143}{250}\sum_{i\in\mathcal{I}} f(A', A'_{\mathcal{I}\setminus\{i\}}).$$

By combining the previous inequality with the bound $\sum_{r\in R} f(A', A'_{\mathcal{I}\setminus\{r\}}) \ge 29/50 \sum_{i\in\mathcal{I}} f(A', A'_{\mathcal{I}\setminus\{i\}})$, we have $\sum_{r\in R, r \text{ is good}} f(A', A'_{\mathcal{I}\setminus\{r\}}) \ge \frac{1}{125}\sum_{i\in\mathcal{I}} f(A', A'_{\mathcal{I}\setminus\{i\}})$.

$\square$

**Lemma 3.8.** *Let $A' \in \mathbb{R}^{n\times d}$ and $S \in \mathbb{R}^{k\times d}$ be the input matrices for Algorithm 2, where $k$ is a positive integer and $S$ is the solution to the CSS problem on $A'$. Suppose that $f(A', S) \ge 25(k+1) \cdot f_k(A', opt)$. Then, with probability at least $1/1375$, Algorithm 2 returns a new solution $S'$ with $f(A', S') \le (1 - 1/(100k))f(A', S)$.*

*Proof.* Let $\mathcal{I}$ denote the set of column indices in $S$. Let $S^*$ be the optimal solution with exactly $k$ columns selected, and let $\mathcal{I}^*$ be set of the columns indices in $S^*$. Let $p$ denote the column index obtained by the steps 2-3 of Algorithm 2. Let $p^* \in \mathcal{I}^*$ be the column corresponding to index $p$, as

determined by the mapping function $\phi(\cdot)$. According to Lemma 3.6, the probability of sampling a good column index is at least

$$\frac{(1/125)\sum_{h\in H, h \text{ is good}} f(A', A'_{\mathcal{I}\setminus\{h\}})}{\sum_{i\in\mathcal{I}} f(A', A'_{\mathcal{I}\setminus\{i\}})} \geq \frac{1}{125}.$$

Let $h$ represent an arbitrary good column index. By Lemma 3.4 and the non-increasing property of residual error $f(\cdot)$ (as shown in the proof of Lemma 3.6), we have

$$\mathbb{E}[f(A', A'_{\mathcal{I}\cup\{p\}})] \leq f_k(A', opt) + \frac{1}{10}f(A', S)$$

$$\leq f(A', A'_{\mathcal{I}^*\setminus\{h^*\}}) + \frac{1}{10}f(A', S).$$

Hence, by the Markov inequality, we obtain

$$\mathbf{Pr}[f(A', A'_{\mathcal{I}\cup\{p\}}) \leq \frac{11}{10}(f(A', A'_{\mathcal{I}^*\setminus\{h^*\}}) + \frac{1}{10}f(A', S))|h \in H] \geq \frac{1}{11}.$$

Combining the above inequalities, the probability that the sampled column index $p$ replaces some good column index $h$ is at least

$$\frac{1}{125}\cdot\frac{1}{11} > \frac{1}{1375}.$$

Similarly, according to Lemma 3.7, the probability of sampling a good column index is at least

$$\frac{(1/125)\sum_{r\in R, r \text{ is good}} f(A', A'_{\mathcal{I}\setminus\{r\}})}{\sum_{i\in\mathcal{I}} f(A', A'_{\mathcal{I}\setminus\{i\}})} \geq \frac{1}{125}.$$

By Lemma 3.4 and the Markov inequality, we obtain

$$\mathbf{Pr}[f(A', A'_{\mathcal{I}\cup\{p\}}) \leq \frac{11}{10}(f(A', A'_{\mathcal{I}^*\setminus\{r^*\}}) + \frac{1}{10}f(A', S))|r \in R] \geq \frac{1}{11}.$$

Thus, the probability that the sampled column index $p$ for some good column index $r$ is at least

$$\frac{1}{125}\cdot\frac{1}{11} > \frac{1}{1375}.$$

By Definition 3.1, we have $\tau(A', A'_{\mathcal{I}\cup\{p\}}, \mathcal{I}\cup\{p\}\setminus\{q\}) = f(A', A'_{\mathcal{I}\cup\{p\}\setminus\{q\}}) - f(A', A'_{\mathcal{I}\cup\{p\}})$, and $\tau(A', S, \mathcal{I}\setminus\{q\}) = f(A', A'_{\mathcal{I}\setminus\{q\}}) - f(A', S)$.

Now, for the column index $q \in H\cup R$, we can upper bound the updated residual error $f(A', S')$ as follows.

$$f(A', S') \leq f(A', A'_{\mathcal{I}\setminus\{q\}\cup\{p\}})$$
$$= f(A', S) - \left(f(A', S) - f(A', A'_{\mathcal{I}\setminus\{q\}\cup\{p\}})\right)$$
$$= f(A', S) - \left(f(A', S) - f(A', A'_{\mathcal{I}\cup\{p\}}) + f(A', A'_{\mathcal{I}\cup\{p\}}) - f(A', A'_{(\mathcal{I}\cup\{p\})\setminus\{q\}})\right)$$
$$= f(A', S) - \left(f(A', S) - \tau(A', A'_{\mathcal{I}\cup\{p\}}, (\mathcal{I}\cup\{p\})\setminus\{q\}) - f(A', A'_{\mathcal{I}\cup\{p\}})\right)$$
$$= f(A', S) - (f(A', \mathcal{I}\setminus\{q\}) - \tau(A', S, \mathcal{I}\setminus\{q\})$$
$$- \tau(A', A'_{\mathcal{I}\cup\{p\}}, (\mathcal{I}\cup\{p\})\setminus\{q\}) - f(A', A'_{\mathcal{I}\cup\{p\}}))$$
$$\leq f(A', S) - (f(A', \mathcal{I}\setminus\{q\}) - \tau(A', S, \mathcal{I}\setminus\{q\}) - \tau(A', A'_{\mathcal{I}\cup\{p\}}, (\mathcal{I}\cup\{p\})\setminus\{q\})$$
$$- \frac{11}{10}(f(A', A'_{\mathcal{I}^*\setminus\{q^*\}}) + \frac{1}{10}f(A', S)))$$
$$\leq f(A', S) - \frac{1}{100k}f(A', S).$$

Thus, by combining the two cases, we obtain that with probability at least $1/1375$,

$$f(A', S') \leq (1 - 1/(100k))f(A', S).$$

$\square$

**Lemma 3.10.** *Let $A \in \mathbb{R}^{n \times d}$ be an input matrix, and let $k$ be a positive integer. Define $D$ as an $n \times d$ matrix with elements*

$$D_{ij} = \begin{cases} \frac{\|A - S_1 S_1^\dagger A\|_F}{(52 \min\{n,d\}(k+1)!)^{1/2}}, & if\ i = j \\ 0, & otherwise \end{cases},$$

*where $S_1$ is obtained by executing the first round of steps 3-6 in Algorithm 1. Let $A' = A + D$. The solution $S_2$ returned by executing Algorithm 2 for $T = O(k^2 \log k)$ iterations satisfies*

$$\mathbb{E}[\|A' - S_2 S_2^\dagger A'\|_F^2] \leq 53(k+1)\|A - A_k\|_F^2,$$

*where $A_k$ is the best rank-$k$ approximation for $A$.*

*Proof.* Let $A'_k$ be the best rank-$k$ approximation of $A'$, and $A_k$ be the best rank-$k$ approximation of $A$. For the initial solution $S_1$, Deshpande and Vempala [15] provide an approximation ratio of $(k+1)!$. By Lemma 3.9, we obtain

$$
\begin{aligned}
\mathbb{E}[\|A' - S_2 S_2^\dagger A'\|_F^2] &\leq 26(k+1)\|A' - A'_k\|_F^2 \\
&\leq 26(k+1)\|A' - A_k\|_F^2 \\
&= 26(k+1)\|A' - A + A - A_k\|_F^2 \\
&\leq 52(k+1)\left(\|A' - A\|_F^2 + \|A - A_k\|_F^2\right) \\
&\leq 52(k+1)\left(\|D\|_F^2 + \|A - A_k\|_F^2\right) \\
&= 52(k+1)\left(\|A - A_k\|_F^2 + \sum_{i=1}^{\min\{n,d\}} D_{ii}^2\right) \\
&\leq 52(k+1)\left(\|A - A_k\|_F^2 + \min\{n,d\} \cdot \left(\frac{\|A - S_1 S_1^\dagger A\|_F^2}{52 \min\{n,d\} \cdot (k+1)!}\right)\right) \\
&= 52(k+1)\left(\|A - A_k\|_F^2 + \frac{\|A - S_1 S_1^\dagger A\|_F^2}{52(k+1)!}\right) \\
&\leq 52(k+1)\left(\|A - A_k\|_F^2 + \frac{\|A - A_k\|_F^2}{52}\right) \\
&= 53(k+1)\|A - A_k\|_F^2,
\end{aligned}
$$

where the second inequality holds because $\|A' - A'_k\|_F^2 \leq \|A' - B\|_F^2$ for any $n \times d$ matrix $B$ with $rank(B) = k$, and the third inequality results from the triangle inequality $\|A + B\|_F^2 \leq 2(\|A\|_F^2 + \|B\|_F^2)$ for any two matrices $A, B \in \mathbb{R}^{n \times d}$. $\qquad\square$

## A.2 Complementary Experiments

### A.2.1 Experiments on Small Datasets

In this section, we compare our algorithm with four algorithms (TwoStage, Greedy, VolumeSampling, and ILS) introduced in Section 4 using 14 small real-world datasets. Following [23, 1], these datasets are listed in Table 5 and all datasets can be found on the website[6].

We use the same experimental settings as in Section 4 to run these algorithms on 14 datasets. The running time results are presented in Tables 6 and 7. LSCSS is at least 2 times faster than the Greedy,

---

[6]http://www.cs.columbia.edu/CAVE/software/softlib/coil-20.php.,
http://www.sheffield.ac.uk/eee/research/iel/research/face.,
http://www.cs.nyu.edu/~roweis/data.html.,
http://vision.ucsd.edu/~leekc/ExtYaleDatabase/ExtYaleB.html.,
http://www.iro.umontreal.ca/~lisa/twiki/bin/view.cgi/Public/PublicDatasets.,
https://archive.ics.uci.edu/datasets,
https://www.csie.ntu.edu.tw/~cjlin/libsvmtools/datasets/,
and http://www.cl.cam.ac.uk/research/dtg/attarchive/facedatabase.html.

VolumeSampling, and ILS algorithms. The error ratio, reported as mean $\pm$ std, with the best results highlighted in bold, is shown in Tables 8 and 9. LSCSS outperforms all other algorithms in terms of quality across all datasets.

### A.2.2 Experiments on QR with Column Pivoting and LSCSS Algorithms

In this section, we present experimental results comparing the performance of our proposed LSCSS algorithm with the QR with Column Pivoting (QRP) algorithm [24] for varying $k$. The QRP algorithm uses column pivoting to improve the traditional QR decomposition.

To compare our proposed algorithm with the QRP algorithm, we conducted experiments on five datasets: CMHS, ELD, Gas, FAds, and TGas (listed in Table 2). Due to the memory requirement of the QRP algorithm exceeding 2TB for the remaining datasets in Table 2, those datasets were excluded from our comparison. For the QRP algorithm, we followed the procedure outlined in [24], first obtaining the permutation matrix $P$ that satisfies $AP = QR$. Then, we selected the top $k$ elements from the diagonal of $P$ as the indices $C$ of the $k$ columns and computed the error ratio of solution $A_C$. The detailed results in Table 10 show that our algorithm achieves lower error ratios across all datasets and is faster than the QRP algorithm.

Table 4: Comparison results on error ratio for varying $k$ on datasets. If an algorithm fails to output a solution within 48 hours, the error ratio is marked as ">48h". If its memory usage exceeds 2TB, the error ratio is marked as "OOM" (Out of Memory).

| Dataset | k | TwoStage | Greedy | ILS | LSCSS |
|---|---|---|---|---|---|
| CMHS | 5 | 56.9579±1.4638 | 1.7446 | 1.2494±0.0492 | **1.1348±0.0202** |
| | 10 | 254.7578±8.3753 | 1.9111 | 1.4803±0.0380 | **1.2237±0.0894** |
| | 15 | 523.9830±28.9132 | 2.1692 | 1.6455±0.0106 | **1.6134±0.0177** |
| | 20 | 655.8496±13.7001 | 1.9725 | 1.6582±0.0120 | **1.6489±0.0482** |
| | 25 | 815.5154±12.7724 | 1.9762 | 1.7237±0.0251 | **1.6652±0.0628** |
| | 30 | 955.9324±13.1531 | 1.9109 | 1.7012±0.0081 | **1.6377±0.0136** |
| | 50 | 1408.5406±10.4335 | 1.7384 | 1.6851±0.0095 | **1.6250±0.0553** |
| | 100 | 2009.6568±25.6895 | 1.5354 | 1.5681±0.0053 | **1.5042±0.0199** |
| FAds | 5 | 1.1791±0.0063 | 1.0807 | 1.0660±0.0165 | **1.0648±0.0194** |
| | 10 | 1.2094±0.0172 | 1.1026 | 1.0719±0.0002 | **1.0748±0.0208** |
| | 15 | 1.2007±0.0511 | 1.1217 | 1.0727±0.0083 | **1.0726±0.0114** |
| | 20 | 1.2271±0.0082 | 1.1405 | 1.1174±0.0053 | **1.0815±0.0126** |
| | 25 | 1.2598±0.0036 | 1.1477 | 1.1291±0.0431 | **1.1020±0.0090** |
| | 30 | 1.2728±0.0053 | 1.1647 | 1.1011±0.0045 | **1.0898±0.0097** |
| | 50 | 1.3320±0.0068 | 1.2205 | 1.2361±0.0341 | **1.2140±0.0112** |
| | 100 | 1.4565±0.0118 | 1.3274 | 1.3516±0.0522 | **1.2628±0.0195** |
| ELD | 5 | 2.0909±0.0428 | 1.5557 | 1.1626±0.0108 | **1.1570±0.0347** |
| | 10 | 2.4693±0.3801 | 1.6769 | 1.2228±0.0090 | **1.2197±0.0609** |
| | 15 | 2.7804±0.1867 | 1.6912 | 1.2466±0.0157 | **1.2397±0.0384** |
| | 20 | 3.4629±0.1021 | 1.9075 | 1.3616±0.0122 | **1.3580±0.0378** |
| | 25 | 4.6062±0.1384 | 1.9911 | 1.4347±0.0095 | **1.4146±0.0023** |
| | 30 | 5.6617±0.1601 | 2.1623 | 1.4855±0.0312 | **1.4671±0.0634** |
| | 50 | 12.0868±0.1241 | 2.4405 | 1.5643±0.0353 | **1.5475±0.0604** |
| | 100 | 42.7583±0.2457 | 2.9438 | 1.8116±0.0075 | **1.7935±0.0445** |
| TGas | 5 | 4.3402±0.0298 | 1.4203 | 1.2641±0.0115 | **1.1657±0.1101** |
| | 10 | 10.3151±0.0233 | 1.9085 | **1.4812±0.0388** | 1.4858±0.0731 |
| | 15 | 17.2847±0.0530 | 2.0486 | 1.6572±0.0162 | **1.6107±0.0547** |
| | 20 | 25.3236±0.0474 | 2.2949 | 1.6480±0.0220 | **1.5986±0.0879** |
| | 25 | 35.4978±0.0085 | 2.4922 | 1.7614±0.0361 | **1.7588±0.0850** |
| | 30 | 47.0122±0.0811 | 2.4077 | 1.7814±0.0129 | **1.7680±0.1291** |
| | 50 | 109.0546±0.0506 | 2.5511 | **1.7567±0.0185** | 1.7628±0.0938 |
| | 100 | 425.7642±0.0372 | 2.4501 | 1.7955±0.0071 | **1.7873±0.0425** |
| Gas | 5 | 5.4063±0.0635 | 1.5689 | **1.4692±0.0102** | 1.4721±0.0765 |
| | 10 | 12.1241±0.0542 | 1.7191 | **1.6262±0.0760** | 1.6315±0.0747 |
| | 15 | 19.2035±0.0078 | 1.7675 | 1.7093±0.0500 | **1.6779±0.0448** |
| | 20 | 26.1705±0.1036 | 1.7616 | 1.6635±0.0678 | **1.6263±0.0263** |
| | 25 | 34.1813±0.1193 | 1.7907 | 1.6576±0.0365 | **1.6545±0.0031** |
| | 30 | 41.0833±0.3698 | 1.8155 | 1.6540±0.0012 | **1.6346±0.0199** |
| | 50 | 76.6821±1.8847 | 1.8207 | 1.6749±0.0403 | **1.6717±0.0082** |
| | 100 | 209.5885±7.0179 | 2.0473 | 1.6823±0.0196 | **1.6676±0.0165** |
| YaleB | 5 | OOM | 1.6582 | 1.4791±0.0473 | **1.4253±0.0905** |
| | 10 | OOM | 1.7959 | **1.5023±0.0315** | 1.5346±0.0448 |
| | 15 | OOM | >48h | 1.5861±0.0864 | **1.5809±0.0529** |
| | 20 | OOM | >48h | 1.6136±0.0163 | **1.6109±0.0417** |
| | 25 | OOM | >48h | 1.6422±0.0878 | **1.6341±0.0301** |
| | 30 | OOM | >48h | >48h | **1.6745±0.0365** |
| | 50 | OOM | >48h | >48h | **1.7484±0.0505** |
| | 100 | OOM | >48h | >48h | **1.8616±0.0954** |
| Mnist8m | 5 | 2.3392±0.0009 | 1.7131 | **1.1239±0.0094** | 1.1277±0.0494 |
| | 10 | 3.0365±0.0017 | 1.9146 | 1.2357±0.0158 | **1.1451±0.0188** |
| | 15 | >48h | >48h | 1.2627±0.0269 | **1.2380±0.0480** |
| | 20 | >48h | >48h | >48h | **1.3350±0.0730** |
| | 25 | >48h | >48h | >48h | **1.3659±0.0769** |
| | 30 | >48h | >48h | >48h | **1.3697±0.0877** |
| | 50 | >48h | >48h | >48h | **1.6109±0.0608** |
| | 100 | >48h | >48h | >48h | **1.8554±0.0401** |
| Epsilon | 5 | 1.1391±0.0483 | 1.0203 | 1.0818±0.0413 | **1.0190±0.0105** |
| | 10 | 1.0774±0.0082 | 1.0243 | 1.0921±0.0281 | **1.0241±0.0141** |
| | 15 | 1.0805±0.0005 | 1.0227 | 1.0822±0.0229 | **1.0227±0.0080** |
| | 20 | 1.0876±0.0051 | 1.0223 | 1.0814±0.0147 | **1.0220±0.0165** |
| | 25 | 1.0884±0.0002 | 1.0223 | 1.0850±0.0094 | **1.0220±0.0011** |
| | 30 | 1.0915±0.0079 | 1.0226 | 1.0742±0.0112 | **1.0224±0.0067** |
| | 50 | 1.0829±0.0004 | 1.0246 | 1.0730±0.0093 | **1.0242±0.0125** |
| | 100 | 1.0826±0.0019 | 1.0298 | 1.0674±0.0081 | **1.0554±0.0467** |

Table 5: Summary of 14 Small Datasets

| Dataset | Rows | Columns | Dataset | Rows | Columns |
|---|---|---|---|---|---|
| sonar | 208 | 60 | mediamill | 30993 | 120 |
| BinaryAlpha | 1404 | 320 | musk | 7074 | 168 |
| dna | 2000 | 180 | arrhythmia | 452 | 279 |
| CTs | 53500 | 386 | sEMG | 1800 | 2500 |
| ORL | 400 | 10304 | USPS | 9298 | 256 |
| COIL20 | 1440 | 1024 | ISOLET | 7797 | 617 |
| mnist | 10000 | 784 | UMIST | 575 | 12880 |

Table 6: Comparison results of running time for varying $k$ on 6 small datasets. If an algorithm fails to output a solution within 48 hours, the running time is set as ">48h".

| Dataset | k | TwoStage | Greedy | VolumeSampling | ILS | LSCSS |
|---|---|---|---|---|---|---|
| arrhythmia | 5 | 1.14 | 1.70 | 71.75 | 0.32 | **0.02** |
| | 10 | 1.13 | 1.53 | 146.81 | 1.10 | **0.05** |
| | 15 | 1.16 | 2.37 | 217.47 | 2.77 | **0.07** |
| | 20 | 1.18 | 3.32 | 241.11 | 5.72 | **0.14** |
| | 25 | 1.19 | 4.15 | 282.61 | 8.47 | **0.22** |
| | 30 | 1.17 | 5.02 | 294.37 | 10.96 | **0.28** |
| | 50 | 1.23 | 8.38 | 482.52 | 39.51 | **0.67** |
| | 100 | 1.45 | 18.44 | 963.81 | 190.94 | **1.02** |
| binaryalpha | 5 | 1.19 | 1.09 | 57.28 | 0.33 | **0.04** |
| | 10 | 1.26 | 2.21 | 112.59 | 1.78 | **0.10** |
| | 15 | 1.29 | 3.31 | 162.58 | 4.25 | **0.16** |
| | 20 | 1.29 | 4.68 | 204.88 | 7.90 | **0.28** |
| | 25 | 1.32 | 5.87 | 260.07 | 12.03 | **0.36** |
| | 30 | 1.31 | 7.89 | 297.84 | 15.68 | **0.51** |
| | 50 | 1.38 | 14.29 | 487.54 | 50.88 | **0.75** |
| | 100 | 1.56 | 27.69 | 929.32 | 254.56 | **1.39** |
| COIL20 | 5 | 1.57 | 3.84 | 1498.82 | 1.09 | **0.03** |
| | 10 | 1.85 | 8.34 | 3050.59 | 4.26 | **0.16** |
| | 15 | 2.03 | 13.56 | 4261.67 | 9.77 | **0.28** |
| | 20 | 2.07 | 18.43 | 6147.76 | 17.20 | **0.44** |
| | 25 | 2.02 | 24.72 | 7242.89 | 28.77 | **0.49** |
| | 30 | 2.03 | 30.44 | 9834.91 | 38.89 | **0.76** |
| | 50 | 2.18 | 50.90 | 17081.15 | 113.82 | **1.59** |
| | 100 | 2.49 | 89.30 | 73120.66 | 502.32 | **2.37** |
| CTS | 5 | 13.18 | 20.79 | 5119.43 | 10.30 | **0.97** |
| | 10 | 15.37 | 46.10 | 10639.06 | 37.18 | **2.63** |
| | 15 | 15.59 | 72.65 | 14595.50 | 81.18 | **3.74** |
| | 20 | 15.42 | 101.63 | 23074.20 | 149.07 | **6.11** |
| | 25 | 15.45 | 133.56 | 26595.60 | 237.15 | **7.65** |
| | 30 | 15.58 | 166.70 | 31913.35 | 351.73 | **8.28** |
| | 50 | 15.71 | 317.13 | 51880.07 | 1120.75 | **14.36** |
| | 100 | **17.76** | 841.74 | 104368.19 | 6138.42 | 38.39 |
| dna | 5 | 0.41 | 0.43 | 23.34 | 0.57 | **0.04** |
| | 10 | 0.48 | 1.08 | 46.23 | 3.01 | **0.08** |
| | 15 | 0.43 | 1.67 | 86.54 | 6.38 | **0.17** |
| | 20 | 0.53 | 2.32 | 101.05 | 11.31 | **0.24** |
| | 25 | 0.54 | 3.17 | 117.49 | 18.24 | **0.32** |
| | 30 | 0.52 | 3.88 | 151.58 | 26.98 | **0.49** |
| | 50 | 0.67 | 6.24 | 253.12 | 72.95 | **0.53** |
| | 100 | **0.97** | 567.23 | 960.52 | 400.13 | 1.07 |
| ISOLET | 5 | 2.04 | 10.39 | 3617.83 | 3.79 | **0.35** |
| | 10 | 2.29 | 22.89 | 5340.22 | 13.78 | **0.73** |
| | 15 | 2.24 | 36.11 | 7067.92 | 29.61 | **1.04** |
| | 20 | 2.25 | 49.81 | 8483.23 | 51.92 | **1.32** |
| | 25 | 2.23 | 64.70 | 18531.69 | 82.38 | **1.81** |
| | 30 | 2.24 | 79.76 | 21504.78 | 120.58 | **2.05** |
| | 50 | 2.39 | 145.43 | 37048.08 | 343.95 | **2.28** |
| | 100 | **2.62** | 333.87 | 52324.39 | 1512.94 | 3.92 |

Table 7: Comparison results on running time for varying $k$ on 8 small datasets. If an algorithm fails to output a solution within 48 hours, the running time is set as ">48h".

| Dataset | k | TwoStage | Greedy | VolumeSampling | ILS | LSCSS |
|---|---|---|---|---|---|---|
| mediamill | 5 | 0.64 | 4.75 | 1739.68 | 4.93 | **0.52** |
| | 10 | 4.63 | 10.11 | 2308.06 | 18.13 | **1.06** |
| | 15 | 4.51 | 15.57 | 2739.37 | 40.99 | **1.86** |
| | 20 | 6.55 | 21.39 | 3658.98 | 72.93 | **2.43** |
| | 25 | 7.54 | 27.41 | 4576.99 | 104.55 | **3.53** |
| | 30 | 10.62 | 33.53 | 5481.76 | 143.93 | **4.26** |
| | 50 | 13.78 | 60.84 | 9143.62 | 467.94 | **8.85** |
| | 100 | 21.23 | 114.71 | 17427.80 | 2727.86 | **15.41** |
| sonar | 5 | 0.19 | 0.05 | 0.71 | 0.02 | **0.01** |
| | 10 | 0.16 | 0.10 | 1.06 | 0.07 | **0.02** |
| | 15 | 0.16 | 0.14 | 1.86 | 0.15 | **0.03** |
| | 20 | 0.17 | 0.18 | 2.55 | 0.26 | **0.04** |
| | 25 | 0.15 | 0.22 | 2.97 | 0.41 | **0.07** |
| | 30 | 0.16 | 0.25 | 3.43 | 0.63 | **0.11** |
| | 50 | 0.29 | 0.34 | 5.39 | 1.90 | **0.18** |
| musk | 5 | 0.72 | 2.02 | 10.37 | 1.39 | **0.10** |
| | 10 | 0.75 | 4.65 | 19.89 | 5.77 | **0.15** |
| | 15 | 0.76 | 7.28 | 29.05 | 14.05 | **0.22** |
| | 20 | 0.79 | 10.17 | 39.21 | 24.34 | **0.34** |
| | 25 | 0.81 | 13.15 | 47.87 | 41.61 | **0.40** |
| | 30 | 0.87 | 16.27 | 55.09 | 57.71 | **0.68** |
| | 50 | 0.83 | 29.45 | 90.17 | 184.35 | **0.75** |
| | 100 | **1.25** | 60.38 | 178.53 | 761.19 | 1.90 |
| sEMG | 5 | 4.99 | 40.26 | 6290.13 | 3.28 | **0.45** |
| | 10 | 5.38 | 81.72 | 14800.67 | 10.92 | **1.04** |
| | 15 | 5.98 | 124.24 | 31659.17 | 21.72 | **1.51** |
| | 20 | 7.04 | 166.41 | 63172.23 | 38.54 | **2.02** |
| | 25 | 7.29 | 208.73 | >48h | 57.43 | **2.47** |
| | 30 | 7.71 | 253.16 | >48h | 79.04 | **3.28** |
| | 50 | 8.26 | 446.44 | >48h | 221.22 | **4.52** |
| | 100 | 10.83 | 1003.05 | >48h | 920.75 | **8.79** |
| ORL | 5 | 2.88 | 26.40 | 4670.13 | 3.37 | **0.21** |
| | 10 | 3.64 | 53.72 | 8317.94 | 8.18 | **0.27** |
| | 15 | 9.04 | 80.43 | 12421.25 | 14.23 | **0.56** |
| | 20 | 23.67 | 108.34 | 16367.79 | 21.98 | **0.78** |
| | 25 | 33.4 | 138.25 | 20450.05 | 31.43 | **1.13** |
| | 30 | 35.64 | 168.26 | 24341.61 | 43.89 | **1.25** |
| | 50 | 37.32 | 294.22 | 45980.34 | 107.87 | **2.54** |
| | 100 | 47.36 | 716.27 | 96733.46 | 386.42 | **7.91** |
| UMIST | 5 | 2.47 | 69.85 | 10852.31 | 6.02 | **0.2** |
| | 10 | 2.94 | 140.38 | 21398.78 | 14.37 | **0.53** |
| | 15 | 7.52 | 212.42 | 31965.75 | 25.03 | **0.90** |
| | 20 | 22.61 | 278.14 | 44963.09 | 38.73 | **1.13** |
| | 25 | 31.67 | 328.22 | 53477.56 | 53.96 | **1.54** |
| | 30 | 31.95 | 404.45 | 71528.05 | 69.63 | **1.80** |
| | 50 | 33.31 | 732.53 | >48h | 168.18 | **4.11** |
| | 100 | 39.54 | 1532.27 | >48h | 428.62 | **11.97** |
| mnist | 5 | 3.32 | 22.45 | 2834.30 | 4.73 | **0.38** |
| | 10 | 3.36 | 47.99 | 5582.86 | 17.21 | **0.8** |
| | 15 | 3.41 | 71.95 | 8039.87 | 37.67 | **1.33** |
| | 20 | 3.49 | 96.56 | 10681.44 | 68.24 | **1.88** |
| | 25 | 3.45 | 124.43 | 13936.23 | 104.85 | **2.43** |
| | 30 | 3.52 | 153.30 | 19584.82 | 154.39 | **3.23** |
| | 50 | **3.69** | 267.98 | 27634.12 | 435.65 | 4.71 |
| | 100 | **4.16** | 572.09 | 58759.88 | 1907.21 | 7.84 |
| USPS | 5 | 0.91 | 1.15 | 303.02 | 2.24 | **0.12** |
| | 10 | 1.14 | 2.25 | 628.31 | 8.11 | **0.25** |
| | 15 | 1.13 | 3.35 | 947.55 | 18.84 | **0.33** |
| | 20 | 1.12 | 4.60 | 1307.11 | 34.29 | **0.66** |
| | 25 | 1.29 | 6.06 | 1584.26 | 54.62 | **0.82** |
| | 30 | 1.25 | 7.59 | 1899.81 | 77.41 | **1.20** |
| | 50 | **1.56** | 15.26 | 3327.07 | 241.06 | 2.37 |
| | 100 | **2.21** | 41.90 | 6458.70 | 1088.98 | 6.79 |

Table 8: Comparison results on error ratio for varying $k$ on 6 small datasets. If an algorithm fails to output a solution within 48 hours, the running time is set as ">48h".

| Dataset | k | TwoStage | Greedy | VolumeSampling | ILS | LSCSS |
|---------|---|----------|--------|----------------|-----|-------|
| CTS | 5 | 17.3085±1.1164 | 1.1714 | 1.4125±0.0905 | 1.1805±0.0062 | **1.1682±0.0236** |
| | 10 | 17.9368±1.3967 | 1.2812 | 1.5634±0.0785 | 1.3079±0.0013 | **1.2658±0.0208** |
| | 15 | 16.9421±0.8017 | **1.3156** | 1.5160±0.0340 | 1.4088±0.0470 | 1.3688±0.0207 |
| | 20 | 20.3040±0.4101 | **1.3452** | 1.5497±0.0613 | 1.4070±0.0320 | 1.3777±0.0225 |
| | 25 | 18.3049±0.3146 | 1.3677 | 1.5951±0.0837 | 1.3853±0.0065 | **1.3648±0.0317** |
| | 30 | 18.2682±5.9413 | 1.3834 | 1.6504±0.0167 | 1.3929±0.0009 | **1.3694±0.0182** |
| | 50 | 1.4661±0.0712 | 1.4342 | 1.6428±0.0550 | 1.4568±0.0029 | **1.4175±0.0174** |
| | 100 | 1.5302±0.0429 | 1.5070 | 1.7487±0.0286 | 1.7778±0.0996 | **1.5048±0.0064** |
| musk | 5 | 1.4280±0.0404 | **1.1746** | 1.5812±0.0328 | 1.2011±0.0015 | **1.1746±0.0855** |
| | 10 | 1.4429±0.0841 | 1.2872 | 1.7471±0.0585 | 1.3196±0.0015 | **1.2803±0.0816** |
| | 15 | 1.7174±0.0917 | 1.4085 | 1.8071±0.1319 | 1.4415±0.0081 | **1.3821±0.0546** |
| | 20 | 1.7424±0.0380 | 1.5289 | 1.8776±0.1250 | 1.5703±0.0141 | **1.4754±0.0312** |
| | 25 | 1.7630±0.0251 | 1.5935 | 1.9104±0.0782 | 1.6437±0.0075 | **1.5439±0.0456** |
| | 30 | 1.8151±0.0754 | 1.6808 | 2.1968±0.1437 | 1.6559±0.0125 | **1.6111±0.0619** |
| | 50 | 2.0740±0.0139 | 1.9803 | 2.4366±0.2390 | 1.8774±0.0172 | **1.8673±0.0401** |
| | 100 | 2.0910±0.0254 | 2.1873 | 3.1927±0.1849 | 2.0254±0.0318 | **1.8991±0.0443** |
| sonar | 5 | 1.4215±0.0306 | 1.3729 | 1.5739±0.0107 | 1.3251±0.0000 | **1.3052±0.0524** |
| | 10 | 1.4578±0.0508 | 1.4410 | 1.6541±0.0624 | 1.4535±0.0239 | **1.3909±0.0432** |
| | 15 | 1.4792±0.0195 | 1.4766 | 1.7626±0.0298 | 1.4845±0.0204 | **1.4236±0.0664** |
| | 20 | 1.7213±0.0801 | 1.5664 | 1.8673±0.0389 | 1.5285±0.0108 | **1.4738±0.0348** |
| | 25 | 1.6245±0.0208 | 1.6982 | 1.9229±0.1503 | 1.6288±0.0245 | **1.5287±0.0421** |
| | 30 | 1.7561±0.0430 | 1.8787 | 2.0996±0.0675 | 1.8004±0.0361 | **1.6237±0.0586** |
| | 50 | 2.7735±0.0262 | 2.9159 | 2.4561±0.1478 | 2.7414±0.0215 | **2.3819±0.1598** |
| ORL | 5 | 2.8872±0.0261 | 1.2600 | 1.6455±0.0081 | 1.2904±0.0286 | **1.2547±0.0728** |
| | 10 | 2.8216±0.0359 | **1.3135** | 1.6553±0.0115 | 1.3544±0.0182 | 1.3268±0.0392 |
| | 15 | 1.8991±0.0170 | 1.3453 | 1.6232±0.0650 | 1.4237±0.0091 | **1.3438±0.0206** |
| | 20 | 1.6448±0.0189 | 1.3808 | 1.6555±0.0405 | 1.4900±0.0080 | **1.3684±0.0175** |
| | 25 | 1.5535±0.0085 | 1.4079 | 1.7271±0.0305 | 1.5004±0.0049 | **1.3942±0.0333** |
| | 30 | 1.5677±0.0003 | 1.4263 | 1.7528±0.0836 | 1.5156±0.0066 | **1.4241±0.0355** |
| | 50 | 1.5957±0.0513 | 1.5325 | 1.7890±0.0531 | 1.6214±0.0039 | **1.4807±0.0192** |
| | 100 | 1.7535±0.0228 | 1.5818 | 1.8557±0.0127 | 1.6968±0.0052 | **1.5700±0.0189** |
| USPS | 5 | 1.3988±0.0167 | 1.2998 | 1.5575±0.0723 | 1.3183±0.0219 | **1.1928±0.0674** |
| | 10 | 1.5953±0.0029 | 1.5151 | 1.7870±0.0417 | 1.5065±0.0214 | **1.4632±0.0558** |
| | 15 | 1.6875±0.0521 | 1.6312 | 2.0675±0.0706 | 1.6489±0.0271 | **1.5790±0.0588** |
| | 20 | 1.8765±0.0090 | 1.7242 | 2.0169±0.0802 | **1.6592±0.0152** | 1.6767±0.0328 |
| | 25 | 1.8218±0.0618 | 1.7965 | 2.0815±0.0139 | 1.7480±0.0297 | **1.7356±0.0413** |
| | 30 | 1.8698±0.0596 | 1.8310 | 2.7416±0.0405 | 1.7546±0.0197 | **1.7512±0.0245** |
| | 50 | 1.8854±0.0029 | 2.0011 | 3.0553±0.0837 | 2.0407±0.0576 | **1.9515±0.0394** |
| | 100 | 2.4488±0.0522 | 2.7733 | 3.4671±0.0813 | 2.6115±0.0457 | **2.4361±0.0307** |
| arrhythmia | 5 | 1.5736±0.0249 | 1.1986 | 1.5832±0.0261 | 1.1963±0.0005 | **1.1618±0.0812** |
| | 10 | 1.9387±0.0235 | 1.2684 | 1.7408±0.0712 | 1.2638±0.0080 | **1.2101±0.0194** |
| | 15 | 2.2904±0.0650 | 1.3649 | 1.7815±0.0451 | 1.3349±0.0043 | **1.3110±0.0678** |
| | 20 | 3.1396±0.1686 | 1.3591 | 1.8522±0.0469 | 1.3600±0.0108 | **1.3415±0.0317** |
| | 25 | 3.3725±0.2329 | **1.3759** | 1.9422±0.0903 | 1.4045±0.0222 | 1.3926±0.0365 |
| | 30 | 4.0875±0.0361 | 1.4424 | 2.1703±0.1079 | 1.4506±0.0126 | **1.4012±0.0479** |
| | 50 | 7.5292±0.5595 | 1.6928 | 2.5196±0.1648 | 1.6651±0.0255 | **1.5918±0.0283** |
| | 100 | 49.6032±3.3052 | 1.4074 | 2.8522±0.7297 | 1.8034±0.2081 | **1.3346±0.0136** |
| mnist | 5 | 1.7082±0.0289 | 1.2929 | 1.5199±0.0485 | 1.2792±0.0064 | **1.2434±0.0305** |
| | 10 | 1.7217±0.0175 | 1.3793 | 1.5594±0.0608 | 1.3565±0.0044 | **1.3465±0.0238** |
| | 15 | 1.7193±0.0121 | 1.4167 | 1.5996±0.0304 | 1.4155±0.0043 | **1.3911±0.0187** |
| | 20 | 1.9977±0.0173 | 1.4410 | 1.6817±0.0653 | 1.4360±0.0041 | **1.4172±0.0187** |
| | 25 | 2.0708±0.0174 | 1.4677 | 1.7307±0.0529 | 1.4742±0.0068 | **1.4475±0.0107** |
| | 30 | 2.3759±0.0348 | 1.4977 | 1.8702±0.0311 | 1.4957±0.0046 | **1.4768±0.0080** |
| | 50 | 2.6145±0.0350 | 1.5554 | 1.8630±0.0184 | 1.5647±0.0053 | **1.5257±0.0106** |
| | 100 | 3.0237±0.0655 | 1.6526 | 1.9803±0.0608 | 1.6579±0.0086 | **1.6263±0.0081** |
| ISOLET | 5 | 1.4971±0.0296 | **1.1589** | 1.5024±0.0115 | 1.2701±0.0004 | 1.1598±0.0398 |
| | 10 | 1.4822±0.0256 | 1.2134 | 1.6256±0.0071 | 1.2612±0.0046 | **1.2126±0.0351** |
| | 15 | 1.4773±0.0078 | 1.2693 | 1.5853±0.0192 | 1.3133±0.0434 | **1.2682±0.0327** |
| | 20 | 1.5587±0.0131 | 1.3294 | 1.7625±0.0577 | 1.3691±0.0198 | **1.3169±0.0361** |
| | 25 | 1.6296±0.0152 | 1.3549 | 1.8025±0.0915 | 1.3961±0.0236 | **1.3415±0.0486** |
| | 30 | 1.5717±0.0092 | 1.3784 | 1.8406±0.0703 | 1.4217±0.0131 | **1.3648±0.0458** |
| | 50 | 1.6364±0.0034 | 1.4751 | 1.9458±0.0450 | 1.4928±0.0159 | **1.4562±0.0135** |
| | 100 | 1.6996±0.0274 | 1.6226 | 1.9489±0.0672 | 1.5936±0.0103 | **1.5788±0.0226** |

Table 9: Comparison results on error ratio for varying $k$ on 8 small datasets. If an algorithm fails to output a solution within 48 hours, the running time is set as ">48h".

| Dataset | k | TwoStage | Greedy | VolumeSampling | ILS | LSCSS |
|---------|---|----------|--------|----------------|-----|-------|
| UMIST | 5 | 1.4697±0.0177 | 1.3284 | 1.5403±0.0073 | 1.3222±0.0107 | **1.2960±0.0107** |
| | 10 | 1.5501±0.0149 | 1.3169 | 1.4947±0.0091 | 1.3341±0.0032 | **1.3090±0.0206** |
| | 15 | 1.5496±0.0126 | 1.3630 | 1.5128±0.0124 | 1.3833±0.0042 | **1.3587±0.0107** |
| | 20 | 1.5371±0.0114 | 1.4275 | 1.5364±0.0025 | 1.4570±0.0052 | **1.4209±0.0125** |
| | 25 | 1.5090±0.0089 | 1.4742 | 1.5883±0.0048 | 1.4927±0.0058 | **1.4615±0.0126** |
| | 30 | 1.5997±0.0334 | 1.5160 | 1.6758±0.0083 | 1.5222±0.0092 | **1.5043±0.0076** |
| | 50 | 1.7048±0.0232 | 1.6346 | >48h | 1.6780±0.0045 | **1.6344±0.0137** |
| | 100 | 1.7600±0.0107 | 1.7321 | >48h | 1.7273±0.0056 | **1.7040±0.0084** |
| binaryalpha | 5 | 1.3510±0.0135 | 1.2534 | 1.3391±0.0097 | 1.2451±0.0184 | **1.2209±0.0205** |
| | 10 | 1.3550±0.0081 | 1.3227 | 1.4320±0.0034 | 1.3067±0.0080 | **1.3036±0.0225** |
| | 15 | 1.4281±0.0013 | 1.3898 | 1.5441±0.0067 | 1.3915±0.0020 | **1.3796±0.0131** |
| | 20 | 1.4393±0.0063 | 1.4378 | 1.6791±0.0183 | 1.4366±0.0082 | **1.4336±0.0226** |
| | 25 | 1.4929±0.0119 | 1.4817 | 1.7525±0.0609 | 1.4625±0.0043 | **1.4597±0.0106** |
| | 30 | 1.5108±0.0224 | 1.5118 | 1.7725±0.0208 | 1.4895±0.0079 | **1.4840±0.0150** |
| | 50 | 1.6041±0.0046 | 1.6082 | 1.8562±0.0257 | 1.5813±0.0088 | **1.5795±0.0140** |
| | 100 | 1.6710±0.0142 | 1.6598 | 1.9473±0.0783 | 1.6474±0.0018 | **1.6379±0.0092** |
| COIL20 | 5 | 3.9687±0.0615 | 1.4194 | 1.6815±0.1039 | 1.3428±0.0264 | **1.3396±0.0450** |
| | 10 | 1.7657±0.0345 | 1.4854 | 1.7398±0.0727 | 1.4537±0.0218 | **1.4387±0.0287** |
| | 15 | 1.7081±0.0139 | 1.5274 | 1.7455±0.0664 | 1.5532±0.0111 | **1.5147±0.0286** |
| | 20 | 1.8870±0.0218 | 1.6024 | 1.8599±0.1043 | 1.5866±0.0196 | **1.5421±0.0208** |
| | 25 | 1.9023±0.0104 | 1.6271 | 2.0515±0.0262 | 1.6066±0.0141 | **1.5638±0.0118** |
| | 30 | 1.8841±0.0119 | 1.6389 | 2.0710±0.1581 | 1.6302±0.0130 | **1.5964±0.0167** |
| | 50 | 2.0428±0.0202 | 1.6892 | 2.1290±0.0549 | 1.6747±0.0140 | **1.6506±0.0167** |
| | 100 | 1.8368±0.0310 | 1.8014 | 3.5690±0.2833 | 1.7837±0.0126 | **1.7498±0.0110** |
| dna | 5 | 1.1438±0.0101 | 1.0818 | 1.1294±0.0011 | **1.0818±0.0000** | 1.0878±0.0166 |
| | 10 | 1.1398±0.0965 | 1.1029 | 1.1443±0.0024 | 1.1060±0.0020 | **1.1025±0.0085** |
| | 15 | 1.1445±0.0158 | 1.1235 | 1.1578±0.0138 | 1.1260±0.0009 | **1.1213±0.0043** |
| | 20 | 1.1643±0.0971 | 1.1434 | 1.1738±0.0046 | 1.1453±0.0007 | **1.1386±0.0055** |
| | 25 | 1.1801±0.0485 | 1.1645 | 1.1857±0.0092 | 1.1649±0.0016 | **1.1603±0.0038** |
| | 30 | 1.2006±0.1036 | 1.1851 | 1.2179±0.0108 | 1.1834±0.0009 | **1.1776±0.0078** |
| | 50 | 1.2839±0.0934 | 1.2766 | 1.3095±0.0055 | 1.2692±0.0005 | **1.2617±0.0034** |
| | 100 | 1.6617±0.0656 | 1.6638 | 1.7365±0.0815 | 1.6383±0.0021 | **1.6066±0.0145** |
| mediamill | 5 | 2.0706±0.0479 | 1.6583 | 2.1393±0.1273 | 1.4696±0.0109 | **1.4173±0.1028** |
| | 10 | 1.8658±0.0083 | 1.7480 | 2.4754±0.2760 | 1.6673±0.0309 | **1.5602±0.1459** |
| | 15 | 2.0647±0.0198 | 1.8526 | 2.6652±0.2078 | 1.7285±0.0221 | **1.6312±0.1056** |
| | 20 | 1.9377±0.0391 | 1.9888 | 2.7980±0.1245 | 1.8613±0.0602 | **1.7563±0.0987** |
| | 25 | 2.0998±0.0527 | 2.0537 | 2.8292±0.0740 | 1.8453±0.0329 | **1.8944±0.1191** |
| | 30 | 2.3301±0.0015 | 2.0571 | 2.9513±0.2368 | 1.8841±0.0350 | **1.8636±0.1155** |
| | 50 | 2.2574±0.0269 | 2.1938 | 3.0384±0.1871 | 2.0635±0.0582 | **1.9441±0.0693** |
| | 100 | 2.7410±0.0133 | 2.6013 | 3.3817±0.1265 | 2.3854±0.0423 | **2.0646±0.1056** |
| sEMG | 5 | 1.0291±0.0003 | 1.0231 | 1.0277±0.0006 | 1.0252±0.0010 | **1.0217±0.0058** |
| | 10 | 1.0477±0.0002 | **1.0322** | 1.0452±0.0024 | 1.0358±0.0036 | 1.0343±0.0094 |
| | 15 | 1.0680±0.0039 | **1.0411** | 1.0817±0.0053 | 1.0445±0.0031 | 1.0430±0.0022 |
| | 20 | 1.0886±0.0001 | 1.0495 | 1.1043±0.0032 | 1.0502±0.0053 | **1.0494±0.0052** |
| | 25 | 1.1079±0.0003 | **1.0572** | >48h | 1.0575±0.0075 | **1.0572±0.0049** |
| | 30 | 1.1386±0.0001 | **1.0647** | >48h | 1.0691±0.0021 | 1.0657±0.0010 |
| | 50 | 1.1117±0.0006 | 1.0922 | >48h | 1.0925±0.0016 | **1.0844±0.0004** |
| | 100 | 1.1496±0.0011 | 1.1450 | >48h | 1.1360±0.0008 | **1.1332±0.0054** |

Table 10: Comparison results on error ratio and running time for QRP and LSCSS algorithms

| Dataset | $k$ | QRP | | LSCSS | |
|---------|-----|-------|-------|-------|------|
| | | ratio | time | ratio | time |
| CMHS | 5 | 4.9384 | 1289.86 | 1.1348±0.0202 | 2.82 |
| | 10 | 3.3580 | 1289.87 | 1.2237±0.0894 | 5.17 |
| | 15 | 5.7633 | 1289.87 | 1.6134±0.0177 | 13.39 |
| | 20 | 3.5838 | 1289.90 | 1.6489±0.0482 | 30.45 |
| | 25 | 3.7979 | 1289.88 | 1.6652±0.0628 | 43.27 |
| | 30 | 4.0852 | 1289.91 | 1.6377±0.0136 | 52.82 |
| | 50 | 4.0925 | 1289.93 | 1.6250±0.0553 | 62.36 |
| | 100 | 4.6109 | 1289.98 | 1.5042±0.0199 | 108.98 |
| ELD | 5 | 1.9681 | 124.93 | 1.1570±0.03478 | 2.18 |
| | 10 | 2.5603 | 124.93 | 1.2197±0.0609 | 3.47 |
| | 15 | 2.6873 | 124.94 | 1.2397±0.0384 | 5.36 |
| | 20 | 2.8871 | 124.94 | 1.3580±0.0378 | 13.93 |
| | 25 | 3.4399 | 124.94 | 1.4146±0.0023 | 22.98 |
| | 30 | 3.5369 | 124.95 | 1.4671±0.0634 | 35.03 |
| | 50 | 4.3966 | 124.97 | 1.5475±0.0604 | 66.27 |
| | 100 | 4.8920 | 125.03 | 1.7935±0.0445 | 89.94 |
| Gas | 5 | 2.2778 | 913.83 | 1.4721±0.0765 | 5.83 |
| | 10 | 2.1567 | 913.60 | 1.6315±0.0747 | 11.67 |
| | 15 | 2.5804 | 913.60 | 1.6779±0.0448 | 19.3 |
| | 20 | 2.4450 | 913.62 | 1.6263±0.0263 | 47.9 |
| | 25 | 2.7521 | 913.60 | 1.6545±0.0031 | 130.11 |
| | 30 | 2.8961 | 913.62 | 1.6346±0.0199 | 322.92 |
| | 50 | 2.8465 | 913.63 | 1.6717±0.0082 | 386.15 |
| | 100 | 3.6717 | 913.65 | 1.6676±0.0165 | 850.01 |
| FAds | 5 | 1.1361 | 6487.57 | 1.0648±0.0194 | 3.4 |
| | 10 | 1.1805 | 6487.52 | 1.0748±0.0208 | 10.58 |
| | 15 | 1.2055 | 6487.52 | 1.0726±0.0114 | 17.65 |
| | 20 | 1.2224 | 6487.54 | 1.0815±0.0126 | 18.27 |
| | 25 | 1.2383 | 6487.55 | 1.1020±0.0090 | 22.29 |
| | 30 | 1.2531 | 6487.65 | 1.0898±0.0097 | 22.76 |
| | 50 | 1.2886 | 6487.62 | 1.2140±0.0112 | 36.04 |
| | 100 | 1.3430 | 6487.74 | 1.2628±0.0195 | 95.67 |
| TGas | 5 | 2.7632 | 1239.08 | 1.1657±0.1101 | 17.15 |
| | 10 | 4.0830 | 1239.06 | 1.4858±0.0731 | 30.28 |
| | 15 | 6.4080 | 1239.07 | 1.6107±0.0547 | 36.99 |
| | 20 | 7.1182 | 1239.07 | 1.5986±0.0879 | 49.9 |
| | 25 | 9.3027 | 1239.08 | 1.7588±0.0850 | 61.36 |
| | 30 | 10.4892 | 1239.10 | 1.7680±0.1291 | 77.93 |
| | 50 | 18.9009 | 1239.16 | 1.7628±0.0938 | 114.23 |
| | 100 | 23.9558 | 1239.42 | 1.7873±0.0425 | 243.89 |

