# OpenReview forum: "Linear Time Approximation Algorithm for Column Subset Selection with Local Search"
_NeurIPS.cc/2024/Conference — NeurIPS 2024 poster_

### Official Review · Reviewer_wwuR · 2024-06-18

**Soundness:** 4
**Presentation:** 3
**Contribution:** 3
**Rating:** 7
**Confidence:** 3

**Summary:**

This paper considers the Column Subset Selection (CSS) problem. In this problem the input is an arbitrary $A\in \mathbb{R}^{n\times d}$ and a positive integer $k$ ($k$ is thought to be much smaller than $\min\{n,d\}$). The goal is to output a subset of $k$ columns of $A$ denoted by $S\in \mathbb{R}^{n\times k}$ such that the residual error $\|A- SS^{\dagger}\|_F^2$ is minimized (here $S^{\dagger}$ is the pseudoinverse of $S$). Previously all except one algorithms for this problem ran in time $\Omega(\min\{nd^2,n^2d\})$ time with $O(k^2)$ approximation factors. One algorithm of Deshpande and Vempala did run in linear time, i.e. $O(nd)$ time but had $(k+1)!\sim k^k$ approximation factor. The contribution of this paper is to design an algorithm that achieves linear $O(nd)$ runtime and has $100(k+1)$ approximation ratio.

**Strengths:**

This paper has many strengths. The first is dramatically improving the previous best approximation ratio which was practically infeasible to a reasonable one for small values of $k$. The second is that the authors show the practical feasibility of the algorithm by demonstrating that it is at least 10 times faster than other algorithms across all datasets they consider.

**Weaknesses:**

I dont see any major weaknesses. One minor weakness is that the experimental section lacks a discussion regarding the parameter settings and experimental setup for the baseline algorithms.

**Questions:**

Can the authors please share the experimental details and parameter settings that are used for implementing the algorithms.

**Limitations:**

Authors have addressed limitations.

---

> ### Author Rebuttal · Authors · 2024-08-07
>
> We thank the reviewer for the positive rating and the thoughtful comments. In the following we address the concerns.
>
> **Question 1. Can the authors please share the experimental details and parameter settings that are used for implementing the algorithms.**
>
> Response: We thank the reviewer for raising this question and apologize for not making it clearer in the paper. Below are the experimental details and parameter settings used in our implementations:
>
> - Algorithm Implementation: The algorithms are implemented using Matlab. For randomized algorithms, we ensured reproducibility by setting random seeds. The implementation details, including the source code, will be made available in a public repository for transparency and reproducibility.
>
> - Algorithms and Parameter Settings: In our experimental evaluation, we consider five distinct algorithms as the following summary:
>   1. The TwoStage algorithm involves two processes: in the first stage, $\Theta(k\log k)$ columns are sampled according to the properties of the top-$k$ right singular vectors of the input matrix. In the second stage, exactly $k$ columns are selected via an RRQR factorization of the subset matrix formed by $\Theta(k\log k)$ columns. The algorithm includes an oversampling parameter $c$ that controls the constant in $\Theta(k\log k)$. We test all integer values of $c$ from 1 to 50 to find the solution with the lowest error. Due to the algorithm involving randomness, we construct 40 submatrices by sampling in the first stage, and then run the RRQR factorization to obtain a solution for each submatrix. Finally, we select the best result from 40 solutions.
>   2. The Greedy algorithm is a deterministic algorithm. The core idea is to iteratively find the column $v$ that minimizes $f(S \cup v, A)$ for the current solution $S$ until $S$ contains exactly $k$ columns. We run the algorithm once for each varying $k$ value and return the solution.
>   3. The VolumeSampling algorithm is a randomized algorithm. It uses a sampling probability proportional to the volume of the parallelepiped spanned by the $k$-column subset. In VolumeSampling, we use an algorithm for computing the characteristic polynomial of an $n \times n$ matrix as described in Section 16.6 of [1].
>   4. The ILS algorithm is a heuristic algorithm. It starts by uniformly selecting $k$ columns to construct an initial solution and then improves the quality of the current solution using a heuristic local search operator. We set the number of iterations to $2k$.
>   5. The LSCSS algorithm, given in Algorithm 1 of this paper, uses two-step mixed sampling and local search methods. We set the number of iterations to $2k$, as in the ILS algorithm.
>
> - Experimental Environment: Experiments are conducted on a machine with 72 Intel Xeon Gold 6230 CPUs and 4TB memory. The operating system is Ubuntu 16.04 LTS. The implementation is done in Matlab 2015, with necessary compatibility settings applied.
>
> - Evaluation Metrics: We used the error ratio $\Vert A - SS^\dagger A \Vert_F^2 / \Vert A - A_k \Vert_F^2$ to evaluate the performance of the algorithms, where $A_k$ is the best rank-$k$ approximation of $A$. The execution time of each algorithm is recorded to compare computational efficiency.
>
> - Experimental Procedure: We test the TwoStage, VolumeSampling, ILS, and LSCSS algorithms on each dataset 10 times to calculate the average error ratio and running time. Since the Greedy algorithm is deterministic, it is tested only once per dataset, with both its error ratio and running time recorded.
>
> - Experimental Results: We divide the experimental results into two parts. The first part, presented in Tables 3 and 4, contains the error ratios and running times for 8 large datasets. The second part, presented in Tables 6 to 12, contains the detailed results for 16 smaller datasets. The experimental results show that our algorithm is at least 10 times faster than other algorithms on large datasets, achieving the best accuracy on most datasets. Additionally, on small datasets, our algorithm is at least 2 times faster than other algorithms and achieves the best accuracy in most cases.
>
> We will include these experimental details and parameter settings in the revised version of the paper.
>
> [1] Bürgisser Peter, et al. Algebraic complexity theory, volume 315 of Grundlehren der Mathematischen Wissenschaften [Fundamental Principles of Mathematical Sciences]. Springer-Verlag, Berlin, 1997. With the collaboration of Thomas Lickteig.

---

> > ### Comment · Reviewer_wwuR · 2024-08-12
> > **Response to rebuttal**
> >
> > Thanks to the authors for the details and for addressing my question. My score remains the same.

---

### Official Review · Reviewer_MkHx · 2024-06-25

**Soundness:** 3
**Presentation:** 2
**Contribution:** 3
**Rating:** 6
**Confidence:** 4

**Summary:**

This paper proposes a new algorithm for the column subset selection problem which combines a local search-type strategy with adaptive sampling techniques to obtain an algorithm running in time linear in $nd$ (i.e. the size of the input matrix) for constant $k$. The resulting solution selects exactly $k$ columns and achieves an approximation ratio at most $O(k)$ against the optimal rank $k$ solution given by the SVD, which is comparable to the best known approximation ratio of $(k+1)$. The main idea is that each iteration of the local search can be made to run in linear time by reducing the number of candidates considered via adaptive sampling.

**Strengths:**

This work achieves a new point in the trade-off between running time and approximation ratio for the problem of column subset selection. Column subset selection is an important and heavily studied problem and thus this work should be of interest to many researchers, especially those working in numerical linear algebra.

**Weaknesses:**

The analysis of the algorithm is highly reminiscent of the adaptive sampling analyses from works such as

https://faculty.cc.gatech.edu/~vempala/papers/relative.pdf

https://dl.acm.org/doi/10.1145/1250790.1250884

This includes, for example, the idea of relating the cost difference from the optimum to a bound on the success probability of adaptive sampling (Lemma 3.8) and then boosting with repetition. The only new point here seems to be to use similar ideas to throw away “bad” columns (i.e. use local search) so that the solution size stays at $k$ rather than growing with each iteration. However, this is still a nice and new idea and requires delicate work to make this go through.

**Questions:**

Questions

- For the problem of row sampling for $\ell_p$ subspace approximation, obtaining tight bounds on the number of rows required for a (1+eps) approximation is an open problem under active research (see, e.g., https://dl.acm.org/doi/10.1145/1250790.1250884). Can the combination of local search and adaptive sampling be used to obtain state of the art results for this problem?

Minor comments

- It would be helpful to move the main theorem Theorem 3.9 earlier so that the formal main result can be seen earlier. In particular, the notion of approximation ratio should be clarified before Table 1, since otherwise it is confusing whether the approximation ratio compares against the best rank k solution or the best subset of k columns.
- The wording of Lemmas 3.7 and 3.8 could be unified for sake of consistency.
- You can remove the section 7 header.

**Limitations:**

The authors have addressed the limitations.

---

> ### Author Rebuttal · Authors · 2024-08-07
>
> We thank the reviewer for the positive rating and the thoughtful comments. In the following we address the concerns.
>
> **Question 1. For the problem of row sampling for $\ell_p$ subspace approximation, obtaining tight bounds on the number of rows required for a (1+eps) approximation is an open problem under active research (see, e.g., https://dl.acm.org/ doi/10.1145/1250790.1250884). Can the combination of local search and adaptive sampling be used to obtain state of the art results for this problem?**
>
> Response: We thank the reviewer for raising this interesting question. For the CSS problem, we utilize a linear-time local search method, overcoming the difficulty of enumerating swap pairs in each local search step. However, we have only provided local search solutions for the CSS problem with the Frobenius norm. For the CSS problem with the $\ell_p$ norm, it is much more challenging to use local search than the Frobenius norm with the following reasons: (1) Lemma 3.3 becomes invalid because its inequality does not hold for the $\ell_p$ norm. (2) Lemma 3.4 becomes invalid because it constructs an approximate relationship between the current solution and the SVD.
>
> As given in [1, 2, 3], $\ell_p$ norm CSS problem is a special type of $\ell_p$ norm subspace approximation. Consequently, using local search for $\ell_p$ subspace approximation has the following difficulties: (1) For the $\ell_p$ subspace approximation problem, how to analyze the relationship between the solution after adding or removing a column and the optimal solution is challenging. (2) How to get a high-quality initial solution is difficult. The above two difficulties are two major obstacles to design an approximation algorithm for $\ell_p$ norm subspace approximation by local search. Thus, designing a (1+eps) approximation algorithm for the subspace approximation is much more challenging, which deserves further study.
>
> [1] Deshpande Amit, and Rameshwar Pratap. On Subspace Approximation and Subset Selection in Fewer Passes by MCMC Sampling. arXiv preprint arXiv:2103.11107 (2021).
>
> [2] Deshpande Amit, and Rameshwar Pratap. One-Pass Additive-Error Subset Selection for $\ell_p$ Subspace Approximation and $(k, p)$-Clustering." Algorithmica 85.10 (2023): 3144-3167.
>
> [3]Mahankali  Arvind V., and David P. Woodruff. Optimal $\ell_1$ column subset selection and a fast PTAS for low rank approximation. Proceedings of the 2021 ACM-SIAM Symposium on Discrete Algorithms (SODA), pages 560-578, 2021.

---

> > ### Comment · Reviewer_MkHx · 2024-08-09
> >
> > Thank you for the rebuttal. I understand the extension to $\ell_p$ subspace approximation would not be obvious. I have increased my score.

---

### Official Review · Reviewer_ANHu · 2024-07-02

**Soundness:** 3
**Presentation:** 2
**Contribution:** 2
**Rating:** 5
**Confidence:** 3

**Summary:**

The Column Subset Selection (CSS) problem aims to select a sub-matrix with $k$ columns from a matrix to minimize the residual error. Previous algorithms often have non-linear running times or higher approximation ratios. This paper proposes a local search-based algorithm with linear running time, utilizing a two-step mixed sampling method to reduce enumerations and a matched swap pair construction to maintain approximation quality. Empirical experiments demonstrate superior performance in quality and speed compared to existing methods, particularly in large datasets.

**Strengths:**

- The proposed new iterative algorithm for column subset selection is simple and sound.

- The theoretical analysis of the algorithm appears novel and solid.

- In experiments, the proposal is generally the fastest one together with competitively small errors.

**Weaknesses:**

- Line 41: What is UG?

- Line 132: Is Lemma 2.1 necessary for the main text?

- Algorithms 1 and 2: It would be better to give the full name of the algorithms.

- Section 3 does not clearly discuss the algorithmic distinction beyond ILS and Greedy.

- The theoretical results of the algorithm are mixed with the algorithmic details, making the paper poorly structured. Additionally, the assumptions are not explicitly listed and lack sufficient discussion.

- Table 4: The standard error of Greedy hasn’t been reported. Besides, it would be better to put Table 4 into the main text and discuss why LSCSS achieves superior approximation compared to the competitors.

- Lines 320 and 321 are repetitive.

**Questions:**

- Algorithm 2 (line 2): How would the numerical performance change if $10k$ columns were replaced with a different number of columns (e.g., $2k$ or even a constant [1])?

- Using the objective function to guide which column to swap out shares a similar spirit with recent advanced algorithms for best-subset selection under linear regression for sparsity-constrained optimization [2, 3]. It would be interesting to discuss these works.

- How should $k$ be selected in practice?

- Table 9: The error ratio of TwoStage on the ComCri dataset is weird. It is exceptionally large when $k=50$ but suddenly reduces to a comparable result when $k=100$.

### Reference

[1] Bahmani, Sohail, Bhiksha Raj, and Petros T. Boufounos. "Greedy sparsity-constrained optimization." The Journal of Machine Learning Research 14.1 (2013): 807-841.

[2] Wang, Zezhi, et al. "Sparsity-Constraint Optimization via Splicing Iteration." arXiv preprint arXiv:2406.12017 (2024).

[3] Zhu, Junxian, et al. "A polynomial algorithm for best-subset selection problem." Proceedings of the National Academy of Sciences 117.52 (2020): 33117-33123.

**Limitations:**

The authors have discussed the limitations in the end of this paper.

---

> ### Author Rebuttal · Authors · 2024-08-07
>
> We thank the reviewer for the positive rating and the thoughtful comments. In the following we address the concerns.
>
> **Weakness 1. Line 41: What is UG?**
>
> Response: We thank the reviewer for this question. ``UG-hard" refers to problems that are as hard as the Unique Games problem, based on the Unique Games Conjecture (UGC). If a problem is UG-hard, it means that there is no efficient algorithm to approximate the problem beyond a certain factor, assuming the UGC is true. This suggests the problem is very difficult to solve unless P=NP or the UGC is disproven.
>
> **Weakness 2. Is Lemma 2.1 necessary for the main text?**
>
> Response: We apologize for the confusion. Lemma 2.1 is used in the proof of Lemma 3.2 to ensure that the third inequality (line 417) holds. Thus, it is necessary.
>
> **Question 1. How would the numerical performance change if $10k$ columns were replaced with a different number of $2k$ columns (e.g., or even a constant [1])?**
>
> Response: We thank the reviewer for asking this question. The choice of selecting $10k$ columns is based on ensuring the probability of sampling a good column in each iteration. If we change it to $2k$ columns, it would lower the probability of successfully finding a good column.
>
> In the following, we provide additional results comparing the selection of $10k$ columns and $2k$ columns for our LSCSS algorithm with varying $k$ values on the 8 datasets (listed in paper). The detailed results, shown in Figure 2 and 3 of the attached PDF, indicate that the accuracy of LSCSS is better with the selection of $10k$ columns than with the selection of $2k$ columns.
>
> **Question 2. Using the objective function to guide which column to swap out shares a similar spirit with recent advanced algorithms for best-subset selection under linear regression for sparsity-constrained optimization [2, 3]. It would be interesting to discuss these works.**
>
> Response: We thank the reviewer for asking this question. For best-subset selection in regression, Wang et al. [2] proposed the ABESS algorithm using the splicing technique. The method uses backward sacrifice and forward sacrifice operations to decide which variables should be swapped. The splicing technique improves the quality of subset selection by minimizing the sum of residual squares and continuously optimizes the objective function value.
>
> For the sparsity-constrained optimization problem, Zhu et al. [3] proposed the SCOPE algorithm. This algorithm generates new candidate sets through local coordinate exchanges and uses the objective value to guide these exchanges, ensuring that the objective value decreases in each iteration.
>
> We use a local search algorithm based on two-step mixed sampling to achieve linear-time solutions for the CSS problem. To avoid $O(dk)$ enumerations of swap pairs, we designed the two-step mixed sampling technique, using the objective function to guide which column to swap out, reducing the enumeration of swap pairs from $O(dk)$ to $k$. This leads to the running time of a single local search to $O(ndk^2)$. By analyzing the sampling probability designed based on the objective function, we can provide theoretical guarantees of the local search.
>
> In summary, both our algorithm and those in [2,3] use some form of objective or loss function to guide the selection and swapping of variables. However, our algorithm differs from the ones in [2, 3]. The methods in [2, 3] use the objective function to guide the swapping process, identifying columns that improve the objective function value. Our algorithm uses the objective function to design the sampling probability, which guides the swapping process to accelerate the running time of local search step. Additionally, we can establish a theoretical relationship between the solution generated by this swapping strategy and the optimal one. Using the strategies from [2, 3] might make it difficult to establish this kind of approximate relationship between solutions.
>
> **Question 3. How should $k$ be selected in practice?**
>
> Response: We thank the reviewer for asking this subtle question. As pointed out in [1,2], the value of $k$ is often treated as a constant input, and the choice of $k$ typically ranges between 1 and 500 in experiments. For all the values between 1 and 500, which one is the best is often hard to decide.
>
> For the medical data analysis application of CSS, Shanab et al. [3] gave that the selection of $k$ is related to the objective function value and correlation measures. However, Shanab et al. [3] did not give the procedure to find an appropriate $k$.
>
> As far as we know, there is no relevant result deciding the choice of $k$ in practice. How to determine the appropriate $k$ value is an interesting problem, and deserves further study.
>
> [1] Christos Boutsidis et al. An improved approximation algorithm for the column subset selection problem. In Proc. 20th Annual ACM-SIAM Symposium on Discrete Algorithms, pages 968-977, 2009.
>
> [2] Venkatesan Guruswami and Ali Kemal Sinop. Optimal column-based low-rank matrix reconstruction. In Proc. 23rd Annual ACM-SIAM Symposium on Discrete Algorithms, pages 1207-1214, 2012.
>
> [3] Shanab, S., et al. Stable bagging feature selection on medical data. Journal of Big Data, 8(1):1-18, 2021.
>
> **Question 4. Table 9: The error ratio of TwoStage on the ComCri dataset is weird. It is exceptionally large when $k=50$ but suddenly reduces to a comparable result when $k = 100$.**
>
> Response: We thank the reviewer for this question. The TwoStage algorithm involves sampling a subset of columns to construct a candidate solution. The ComCri dataset contains 123 columns. When $k = 50$, the algorithm samples a candidate set of 89 columns. Under this case, the accuracy is exceptionally large. When $k = 100$, following the method in [1], the sampled size is larger than 123. Then, the algorithm selects all 123 columns, which makes the subsequent process easier. Thus, the accuracy in such cases reduces to a comparable result.

---

### Official Review · Reviewer_oKb7 · 2024-07-12

**Soundness:** 3
**Presentation:** 3
**Contribution:** 3
**Rating:** 6
**Confidence:** 4

**Summary:**

This paper studies column subset selection. Given a matrix A n*d, how to select a matrix A_S of k columns, which preserves the substance in A? That is, reconstruction error ||A - (A_S A_S#) A||_F should be minimized, where A_S# is the psuedo-inverse of A_S. There are many approximation algorithms for this problem, and it is widely studied in theoretical ML/CS literature. There is a plethora of running-time vs approximation factor guarantee in the literature.

This paper tries to match the SoTA approx. factor with faster running time. Indeed, SoTA algorithm has (k+1)-factor approximation with running time O(n d^2 k). If we wanted linear in n and d, best known algorithm had approx. factor (k+1)! with O(n d k^2) running time. This paper's main contribution is 110 (k+1)-approximation with running time O( n d k^3).

Technically, the main innovation is to show that a constrained version of local search works. Local search is a good algorithm where we initialize a decent solution say the (k+1)! approx. solution, and iteratively swap out a column in the current solution with one not in. This step is expensive, having k*d possible combinations and we need to see which is the best! Multiplying this with # of rounds will yield large running times not-linear in d.

This paper overcomes this by showng that not all swaps are needed to be tried. Indeed, they sample O(10k) possible columns based on some form of random sampling (by looking at the residual importance of various columns), and further uniformly sample one of them at random to swap in. Now, the complexity of testing swaps is only O(k)! The challenge is now to show that the algorithm will not get stuck in bad local optima.

For this authors present a set of "test swaps" based on optimal set of columns and algo set of columns, and show that we will end up sampling one such pair with decent chance, if the current soln is far from optimal.

Finally they run experiments comparing with prior art and show better running time AND quality. Overall solid piece of work and worthy of NeurIPS. The randomized local search idea is good, and the backing with empirical evals is also welcome.

**Strengths:**

Writing is good. Comparion with prior art is also nicely done. The algo is clean and elegant, the idea of randomized local search to improve running time is novel (at least to me, authors should mention if similar ideas appear in other uses of local search).

Good of you to run experiments and validate the niceness of algo!

**Weaknesses:**

The problem is important, though it would be nice for authors to present why "subset" selection is significant as opposed to top-k-SVD. It is a bit unfair to expect a full motivation because this is standard problem, but a bit more explanation for the importance of the problem would be nice.

**Questions:**

Are there practical use-cases of the CSS (where the interpretabiltiy is crucial? as opposed to SVD).
In typical use-cases, do we sample rows (from n like a coreset) or columns (from d like dimension reduction)?

The obj. function f doesn't appear to be defined where it is first used.
Can your algo give better guarantees when allowed to pick more than k columns? How does this compare to prior SoTA in this regime?

Where is the randomness captured in Thm 3.9? Is it in expectation? WHP? The proof has some details but the formal statement should also have it.

**Limitations:**

Yes

---

> ### Author Rebuttal · Authors · 2024-08-07
>
> We thank the reviewer for the positive rating and the thoughtful comments. In the following we address the concerns.
>
> **Question 1. Are there practical use-cases of the CSS (where the interpretabiltiy is crucial? as opposed to SVD). In typical use-cases, do we sample rows (from n like a coreset) or columns (from d like dimension reduction)?**
>
> Response: We thank the reviewer for bringing up this issue. Although both SVD and CSS can be used for dimensionality reduction, they adopt quite different strategies. Specifically, SVD achieves dimensionality reduction by decomposing the matrix into $U\Sigma V^\top$, aiming to capture the low-rank structure of the input matrix. The SVD method generates new features as linear combinations of the original features, resulting in lower interpretability. In contrast, CSS focuses more on the interpretability of the solution, and selects columns based on their importance in performing specific tasks. Since it retains the original columns, CSS often offers better interpretability.
>
> A practical application of CSS is as follows. In remote sensing image classification, as given in  [1], CSS selects a subset of the original features, which have clear physical meanings in the context of remote sensing images. As pointed out in [1], the aim of column subset selection is to find the most informative and distinctive features, which provide better interpretability for specific applications.
>
> Saurabh et al. [2] used sampling method for column selection. Greedy method was applied in [1] for the remote sensing image classification, and Jason et al. [3] also applied greedy method. Sampling methods were used in [4, 5] to deal with both row and column selection. Our proposed LSCSS is a local search method used for dimension reduction, and has advantages over the state-of-the-art methods for accuracy and time, which have potentials for many applications.
>
> [1] Benqin Song, et al. New feature selection methods using sparse representation for one-class classification of remote sensing images. IEEE Geoscience and Remote Sensing Letters 18(10):1761-1765, 2020.
>
> [2] Paul Saurabh, et al. Column selection via adaptive sampling. Advances in neural information processing systems 28, pages 406-414, 2015.
>
> [3] Altschuler Jason, et al. Greedy column subset selection: New bounds and distributed algorithms. Proceedings of the 33rd International Conference on Machine Learning, pages 2539-2548, 2016.
>
> [4] Deshpande Amit, and Luis Rademacher. Efficient volume sampling for row/column subset selection. Proceedings 51th Annual Symposium on Foundations of Computer Science, pages 329-338, 2010.
>
> [5] Frieze Alan, et al. Fast Monte-Carlo algorithms for finding low-rank approximations. Journal of the ACM 51(6):1025-1041, 2004.
>
> **Question 2. The obj. function f doesn't appear to be defined where it is first used. Can your algorithm give better guarantees when allowed to pick more than k columns? How does this compare to prior SoTA in this regime?**
>
> Response: We thank the reviewer for raising the question. The objective function $f$ is defined in line 131 (Section 2). In the revised version, we will make the definition of $f$ clearer.
>
> For the CSS problem, selecting more columns can potentially improve the approximation guarantees. Theoretically, we can prove that selecting more than $k$ columns can lead to a decrease in the objective function value, as follows. Let $S_k$ be the solution with $k$ columns, and let $S_{k+1} = S_k \cup v$ be the solution with one additional column $v$. According to line 422 of this paper, $f(S,A) = \text{tr}(A^\top A) - \Vert S S^\dagger A \Vert_F^2$. Therefore, $f(S_{k+1},A) - f(S_k, A) = \Vert S_k S_k^\dagger A \Vert_F^2 - \Vert S_{k+1} S_{k+1}^\dagger A \Vert_F^2$. From Lemma 1 in [1], we have $\Vert S_k S_k^\dagger A \Vert_F^2 - \Vert S_{k+1} S_{k+1}^\dagger A \Vert_F^2 \le 0$. This implies that the function $f(A,S)$ is monotonically non-increasing. Thus, if the local search outputs a solution $S_k$ with $k$ columns, then by adding a column to form $S_{k+1}$, we can at least guarantee that $f(S_{k+1},A) \le f(S_k, A)$.
>
> To demonstrate that the objective function value decreases as the number of columns selected is larger, we conduct experiments using our proposed LSCSS with varying $k$ values on 8 datasets (listed in paper). We use the objective function $f(S,A) = \Vert A - SS^\dagger A \Vert_F^2$ to measure this property. The detailed results, shown in Figure 1 of the attached PDF, suggest that the objective function value indeed decreases as the number of selected columns increases.
>
> Comparing two algorithms on the same dataset with different numbers of selected columns is unfair. When the same number of columns is selected, the algorithm can be compared with the SoTA.
>
> [1] Altschuler Jason, et al. Greedy column subset selection: New bounds and distributed algorithms. Proceedings of the 33rd International Conference on Machine Learning, pages 2539-2548, 2016.
>
> **Question 3. Where is the randomness captured in Thm 3.9? Is it in expectation? WHP? The proof has some details but the formal statement should also have it.**
>
> Response: We apologize for not clearly describing the role of randomness in our submission. In Theorem 3.9, the solution $S$ returned by Algorithm 1 satisfies $\Vert A-SS^\dagger A\Vert_F^2 \le 110(k+1)\Vert A -A_k\Vert_F^2$ in expectation. In the revised version of the paper, we will make it clearer.

---

### Official Review · Reviewer_6BV1 · 2024-07-14

**Soundness:** 3
**Presentation:** 3
**Contribution:** 3
**Rating:** 7
**Confidence:** 3

**Summary:**

The paper addresses the classical CSS problem, and describes significant
improvements over the current state of the art.
The main idea is to run the randomized
adaptive sampling algorithm for selecting k columns,
and follow it by several iterations of swapping (local search),
that further increase the accuracy. The swapping replaces one of the selected
columns with another unselected column when such swapping reduces the error.

**Strengths:**

The main novelty is in the design of the local search.
The algorithm first selects 10k top norm candidates, and then selects uniformly
at random among those. Even though the analysis does not show their algorithm
to be the most accurate, the experimental results clearly demonstrate
the superiority of this approach.

**Weaknesses:**

* There is no experimental comparison with the QRP (QR with Pivoting) method,
  which is known to show very good results in practice.
  It is significantly faster than your algorithm, as it takes:

  4knd - 2 k^2(d+n) + 4 k^3/3

* The running time of the proposed algorithm is

O(nd k^3 log k)

I do not consider it fast. There are other algorithms that run much faster
(eg Frieze, Kannan and Vempala, JACM 2004, and the above mentioned QRP).

I am also not that impressed by the relative bound of 110(k+1),
as there are other algorithms with 100 times better bounds (specifically k+1).
What is most surprising here is your experimental results.


Some notes:

line 41. CSS is known to be NP-hard.
See "Column subset selection is NP-complete" by Shitov, 2021.

line 42. The comment is incorrect wrt reference 2.

**Questions:**

* Why is there no comparison with the QRP?
* How good is your accuracy compared to the optimal of worst case of (k+1)?

---

> ### Author Rebuttal · Authors · 2024-08-07
>
> We thank the reviewer for the positive rating and the thoughtful comments. In the following we address the concerns.
>
> **Question 1. Why is there no comparison with the QRP (QR with Pivoting) method?**
>
> Response: The main reason that QRP is not compared with our CSS method is because they select columns for different purposes. In the QR decomposition process, QRP chooses the most influence column at each step to maximize the numerical stability (i.e., it selects the column with the largest norm as the pivot column and swaps it with the current one), while CSS selects columns to best represent the input matrix. Many CSS algorithms identify the most representative columns (for dimension reduction) by using objective functions or volume sampling to measure column importance. Even though both QRP and CSS choose a subset of columns, they use them for different objectives. Thus, we have focused our comparisons on those  methods that solve CSS.
>
> Since the subset of columns obtained by QRP can also be viewed as the solution of dimension reduction. For completeness, in the following, we compare our proposed algorithm with the subset of columns obtained by QRP. We conducted experiments on 5 datasets: CMHS, ELD, Gas, FAds, and TGas (listed in Table 2 of our submission). For the QRP method, we first call the algorithm in [2] to obtain the permutation matrix $P$ that satisfies $AP=QR$. Then, we select the top $k$ elements of the diagonal of $P$ as the indices $C$ of the $k$ columns and compute the error ratio of the solution formed by $C$ as mentioned in line 290. The detailed results in Table 1 show that our method achieves the lowest error ratios across all datasets and is faster than the QRP method.
>
> | Dataset | k   | QRP Ratio | QRP Time | LSCSS Ratio          | LSCSS Time |
> | ------- | --- | --------- | -------- | -------------------- | ---------- |
> | CMHS    | 10  | 3.3580    | 1289.87  | 1.1478 $\pm$ 0.1094  | 5.17       |
> | CMHS    | 20  | 3.5838    | 1289.90  | 1.1815 $\pm$ 0.0586  | 30.45      |
> | CMHS    | 30  | 4.0852    | 1289.91  | 1.1988 $\pm$ 0.0772  | 52.82      |
> | CMHS    | 50  | 4.0925    | 1289.93  | 1.2535 $\pm$ 0.0560  | 62.36      |
> | CMHS    | 100 | 4.6109    | 1289.98  | 1.3334 $\pm$ 0.0194  | 108.98     |
> | ELD     | 10  | 2.5603    | 124.93   | 1.2929 $\pm$ 0.0652  | 3.47       |
> | ELD     | 20  | 2.8871    | 124.94   | 1.3656 $\pm$ 0.0225  | 13.93      |
> | ELD     | 30  | 3.5369    | 124.95   | 1.4834 $\pm$ 0.0761  | 35.03      |
> | ELD     | 50  | 4.3966    | 124.97   | 1.4946 $\pm$  0.1086 | 66.27      |
> | ELD     | 100 | 4.8920    | 125.03   | 1.7470 $\pm$ 0.0692  | 89.94      |
> | Gas     | 10  | 2.1567    | 913.60   | 1.6315 $\pm$ 0.0747  | 11.67      |
> | Gas     | 20  | 2.4450    | 913.62   | 1.7466 $\pm$ 0.0542  | 47.90      |
> | Gas     | 30  | 2.8961    | 913.62   | 1.6915 $\pm$ 0.0372  | 322.92     |
> | Gas     | 50  | 2.8465    | 913.63   | 1.8173 $\pm$ 0.0287  | 386.15     |
> | Gas     | 100 | 3.6717    | 913.65   | 1.9362 $\pm$ 0.0344  | 850.01     |
> | FAds    | 10  | 1.1805    | 6487.52  | 1.0748 $\pm$ 0.0208  | 10.58      |
> | FAds    | 20  | 1.2224    | 6487.54  | 1.0815 $\pm$ 0.0126  | 18.27      |
> | FAds    | 30  | 1.2531    | 6487.65  | 1.0898 $\pm$ 0.0097  | 22.76      |
> | FAds    | 50  | 1.2886    | 6487.62  | 1.0853 $\pm$ 0.0087  | 36.04      |
> | FAds    | 100 | 1.3430    | 6487.74  | 1.1334 $\pm$ 0.0085  | 95.67      |
> | TGas    | 10  | 4.0830    | 1239.06  | 1.3696 $\pm$ 0.0731  | 30.28      |
> | TGas    | 20  | 7.1182    | 1239.07  | 1.7932 $\pm$ 0.0764  | 49.90      |
> | TGas    | 30  | 10.4892   | 1239.10  | 1.8677 $\pm$ 0.1050  | 77.93      |
> | TGas    | 50  | 18.9009   | 1239.16  | 2.0565 $\pm$ 0.1475  | 114.23     |
> | TGas    | 100 | 23.9558   | 1239.42  | 2.1499 $\pm$ 0.0883  | 243.89     |
> *Table 1: Comparison results of QRP and LSCSS algorithms*
>
> [1] Quintana-Ortí, Gregorio, Xiaobai Sun, and Christian H. Bischof. A BLAS-3 version of the QR factorization with column pivoting. SIAM Journal on Scientific Computing 19.5 (1998): 1486-1494.
>
> **Question 2. How good is your accuracy compared to the optimal of worst case of (k+1)?**
>
> Response: We thank the reviewer for asking this question. Chierichetti et al. [1] proposed an algorithm to obtain the (k+1) ratio by enumerating all $k$-column subsets in the worst case, with exponential time complexity ($n^k$ subsets), which makes it hard to handle large-scale datasets. Theoretically, our proposed LSCSS achieves a $110(k+1)$-approximation.
>
> To compare the accuracy of our proposed LSCSS with the enumeration algorithm in [1], we follow the setting in [2] and construct a synthetic dataset consisting of a $1000 \times 20$ random matrix $A$, where $A_{i,j}$ are i.i.d. from a normal distribution. We provide additional results comparing our LSCSS algorithm with the enumeration algorithm on synthetic dataset. The detailed results show that the error of our method is only $0.05\\%$ higher than that of the enumeration algorithm.
>
> | k   | Enumeration | LSCSS                |
> | --- | ----------- | -------------------- |
> | 5   | 1.0421      | 1.0421 $\pm$ 0.0002  |
> | 10  | 1.0892      | 1.0898 $\pm$ 0.0001  |
> | 15  | 1.1379      | 1.1390 $\pm$  0.0007 |
> *Table 2: Error ratio of the Enumeration algorithm and LSCSS on a synthetic dataset*
>
> [1] Chierichetti Flavio, et al. Algorithms for $\ell_p $ low-rank approximation. Proceedings of the 34th International Conference on Machine Learning, pages 806-814, 2017.
>
> [2] Paul Saurabh, et al. Column selection via adaptive sampling. Advances in neural information processing systems 28, pages 406-414, 2015.

---

### Author Rebuttal · Authors · 2024-08-07

We thank all the reviewers for the positive ratings and thoughtful comments.

---

### Decision · Program_Chairs · 2024-09-25

**Decision:**

Accept (poster)

**Comment:**

This paper provides a new computationally efficient local search method for column subset selection, a central problem in machine learning and randomized numerical linear algebra. It provides an O(k) factor approximation using exactly k column, improving on prior work that either returned more columns (i.e., gave a bicriteria approximation) or obtained a much worse approximation factor. All reviewers felt the paper made progress on this longstanding problem, so should be accepted to the conference. We recommend that the reviewers strengthen their experimental evaluation by  including the comparison to pivoted QR that they ran during the rebuttal period. We would also recommend comparing against the recent “Randomly pivoted Cholesky” method by Chen et al.